# Rethinking Chain-of-Thought from the Perspective of Self-Training

**Zongqian Wu** [* 1]  **Baoduo Xu** [* 1]  **Ruochen Cui** [1]  **Mengmeng Zhan** [1]  **Xiaofeng Zhu** [1 2]  **Lei Feng** [3]

## Abstract

Chain-of-thought (CoT) reasoning has emerged as an effective approach for activating latent capabilities in LLMs. Interestingly, we observe that both CoT reasoning and self-training share the core objective: iteratively leveraging model-generated information to progressively reduce prediction uncertainty. Building on this insight, we propose a novel CoT framework to improve reasoning performance. Our framework integrates two key components: (i) a task-specific prompt module that optimizes the initial reasoning process, and (ii) an adaptive reasoning iteration module that dynamically refines the reasoning process and addresses the limitations of previous CoT approaches, *i.e.,* over-reasoning and high similarity between consecutive reasoning iterations. Extensive experiments show that the proposed method achieves significant advantages in both performance and computational efficiency. Our code is available at: https://github.com/zongqianwu/ST-COT.

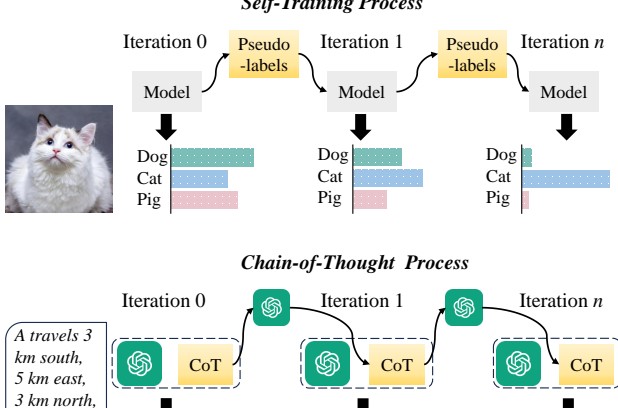

*Figure 1.* Both self-training and CoT reasoning **iteratively** leverage **model-generated information** (pseudo-labels or reasoning processes) to gradually reduce the uncertainty of predictions.

## 1. Introduction

Chain-of-thought (CoT) reasoning has attracted significant attention in recent years due to its capacity to unlock the latent potential of large language models (LLMs) (Wei et al., 2022). By requiring LLMs to explicitly outline intermediate reasoning processes before generating final outputs, CoT effectively improves the reliability of inferences, particularly when tackling complex reasoning tasks.

Previous CoT methods in LLMs can be divided into two categories, *i.e.,* zero-shot CoT (Kojima et al., 2022) and few-shot CoT (Wei et al., 2022). Zero-shot CoT methods rely

on prompts (*e.g.,* "Let's think step by step") to guide LLMs to generate intermediate reasoning processes relevant to the given question, thereby facilitating logical inference. In contrast, few-shot CoT methods provide some examples that include intermediate reasoning processes from the dataset, allowing LLMs to reference these examples during testing to construct reasoning processes. Both zero-shot and few-shot CoT methods leverage the generative capabilities of LLMs to augment question-relevant information, which effectively improves the reliability of inferences. Moreover, this process of information augmentation can be applied iteratively, with deeper iterations enabling LLMs to tackle more complex reasoning tasks (Zhong et al., 2024).

Interestingly, we observe that CoT methods share many conceptual similarities with self-training (a well-established semi-supervised framework), regarding leveraging iteratively model-generated information to enhance task performance (as shown in Figure 1). Concretely, in self-training, pseudo-labels are iteratively generated for unlabeled data and used to retrain the model, thereby progressively enhancing the generalization capabilities of the model. Inspired by this parallel, this paper aims to improve CoT reasoning performance from the perspective of self-training.

---
[*]Equal contribution  [1]School of Computer Science and Engineering, University of Electronic Science and Technology of China, Chengdu, China [2]School of Computer Science and Technology, Hainan University, Haikou, China [3]School of Computer Science and Engineering, Southeast University, Nanjing, China. Correspondence to: Xiaofeng Zhu <seanzhuxf@gmail.com>, Lei Feng <fenglei@seu.edu.cn>.

*Proceedings of the 42nd International Conference on Machine Learning*, Vancouver, Canada. PMLR 267, 2025. Copyright 2025 by the author(s).

To this end, we conduct a theoretical analysis of entropy variation in self-training, which demonstrates that the overall uncertainty of predicted samples exhibits a decreasing trend. Furthermore, our experimental results validate that CoT reasoning follows a similar principle, as it iteratively leverages intermediate reasoning processes to gradually reduce the uncertainty on prediction questions. Based on the insight that CoT reasoning is an uncertainty minimization process, we propose a novel CoT framework to improve reasoning performance, which comprises two main components, *i.e.,* **task-specific prompt** and **adaptive reasoning iteration**. Specifically, the task-specific prompt module is designed to search for optimal prompts with the minimum uncertainty. Unlike generic prompts (*e.g.,* "Let's think step by step"), our tailored prompt effectively guides LLMs to generate initial reasoning processes aligned with the intrinsic characteristics of the task, significantly reducing the number of iterations required for LLMs to obtain correct answers.

After establishing initial CoT reasoning processes, further iterative refinement can help improve the reasoning process. A straightforward method involves integrating the question, reasoning processes, and output into a new input and reusing the prompt for another reasoning round. However, this intuitive method encounters two key challenges: (i) the correct predictions in earlier iterations may turn incorrect after multiple rounds, a phenomenon we term over-reasoning, and (ii) the reasoning generated in new iterations often closely resembles the previous reasoning. To address these challenges, our adaptive reasoning iteration module is implemented to assess the uncertainty of prediction questions. When the uncertainty is low, the current prediction is adopted as the final output, effectively mitigating challenge (i). However, if uncertainty remains high, the reasoning process continues into the next iteration. In these subsequent iterations, we introduce a new prompt and employ reasoning similarity metrics (*e.g.,* the Jaccard index) to guide LLMs in exploring alternative reasoning pathways. By fostering greater diversity across reasoning iterations, we address the challenge (ii), thereby enhancing the ability of LLMs to tackle complex reasoning tasks effectively.

Motivated by the conceptual connection between self-training and chain-of-thought reasoning, our key contributions can be summarized as follows:

- Through a combination of theoretical analysis and experimental validation, we reveal that both self-training and CoT reasoning share the core objective of iteratively leveraging model-generated information to gradually reduce the uncertainty of predictions (*i.e.,* entropy minimization), thereby turning some early incorrect predictions into correct ones.

- We design a task-specific prompt module to search for optimal prompts that generate high-quality initial rea-

soning processes, thereby significantly reducing the number of iterations required for LLMs to obtain correct answers in CoT reasoning.

- we propose an adaptive reasoning iteration module to dynamically refine the CoT reasoning process and address issues of over-reasoning and high similarity between consecutive reasoning iterations.

## 2. Related Work

### 2.1. Self-Training

Self-training (Scudder, 1965) has emerged as a widely used approach in semi-supervised learning, aiming to expand labeled datasets through pseudo-labeling techniques (Yang et al., 2022). An effective pseudo-labeling strategy ensures that the pseudo-labels assigned to unlabeled data align well with the distribution of labeled samples. Recent research has focused on two main directions: selecting high-confidence pseudo-labels and optimizing multi-classifier training frameworks (Amini et al., 2024). For pseudo-label selection, some methods directly incorporate unlabeled samples with high prediction confidence in iterative training (Lee et al., 2013; Zou et al., 2018), while others explore more robust strategies such as majority voting (Bartlett et al., 1998), entropy minimization (Grandvalet & Bengio, 2004), uncertainty estimation (Mukherjee & Awadallah, 2020), noise injection (Miyato et al., 2018), confidence regularization (Zou et al., 2019), and curriculum-based pseudo-labeling (Zhang et al., 2021). However, the presence of incorrect pseudo-labels can mislead model optimization, motivating the development of debiasing techniques and other corrective mechanisms to mitigate their negative impact (Chen et al., 2022).

### 2.2. Chain-of-Thought Reasoning

Trustworthy reasoning is essential for large language models (LLMs), and chain-of-thought (CoT) prompting has emerged as a key technique to improve reliability and transparency by guiding LLMs to generate intermediate reasoning steps (Chu et al., 2024). Early CoT methods relied on implicit prompting to help LLMs solve complex tasks such as arithmetic and commonsense reasoning by decomposing them into simpler subproblems (Chia et al., 2023), demonstrating strong generalization from limited examples and carefully designed prompts (Sun et al., 2023). Recent advances in CoT focus on improving flexibility by generating diverse reasoning paths, particularly for unfamiliar tasks (Kojima et al., 2022; Ling et al., 2024). Methods like PSPrompting (Wang et al., 2023) and Concise-CoT (Nayab et al., 2024) selectively activate pretrained knowledge according to task-specific needs, while approaches such as VerifyCoT (Ling et al., 2024) enhance trustworthiness by generating additional question-answer pairs for validation.

# 3. Understanding Uncertainty in Self-Training and Chain-of-Thought Reasoning

This section investigates the uncertainty in self-training and CoT reasoning by analyzing variations in information entropy and semantic entropy. We observe that while both methods exhibit a general trend of entropy reduction, this pattern does not hold for every data point. Performance improvements are primarily driven by data points where entropy decreases, enabling a transition from incorrect to correct predictions. Specifically, Section 3.1 presents a theoretical analysis of entropy variations in self-training. Building on these insights, Section 3.2 extends the discussion to CoT reasoning, deriving key conclusions that help further enhance the performance of CoT reasoning.

## 3.1. Information Entropy Variation in Self-Training

The primary objective of self-training is to reduce prediction uncertainty by leveraging pseudo-labels generated by the model, where the uncertainty can be quantified using information entropy. As empirically shown in Figure 8(a) in Appendix, the average entropy of predictions across samples progressively declines throughout the iterative training process. This reduction facilitates some samples being corrected from initial mispredictions to accurate predictions.

However, not all samples exhibit the expected reduction in information entropy. This deviation can be attributed to incorrect annotations within pseudo-labels, which may misdirect the model's training direction. To provide deeper insights into the entropy variation of self-training, we conducted a theoretical analysis using a Gaussian mixture model. Specifically, an initial classifier with a small yet constant error was constructed and iteratively updated with pseudo-labels based on model predictions. Through this approach, we examined the classifier's progression from its initial state toward the optimal state, capturing the underlying mechanism of entropy variation. Based on the conclusions of (Frei et al., 2022) regarding the sample complexity of unlabeled samples in self-training, we discover that the intermediate classifier update process is a rotation of the initial classifier towards the Bayes optimal classifier. This conclusion is stated in the following lemma.

**Lemma 3.1.** *Suppose $(x, y) \sim \mathcal{D}$ where $\mathcal{D}$ is a Gaussian mixture models in $\mathbb{R}^d \times \{\pm 1\}$ with mean $\mu$ satisfying $\|\mu\| = \Theta(1)$, i.e., $y \sim \mathrm{Unif}(\{\pm 1\})$ and $x|y \sim \mathcal{N}(y\mu, I)$. Let $\ell(z) = \log(1 + \exp(-z))$, and assume $\sigma \geq \max(1, \|\mu\|)$. Assume we can access a initial classifier $\beta_{\mathrm{init}}$ which satisfies $\mathrm{Pr}_{(x,y)\sim\mathcal{D}}[y \neq \mathrm{sgn}(\beta_{\mathrm{init}}^T x)] = O(1)$. Let $\varepsilon, \delta \in (0, 1)$, and assume that $B = \tilde{\Omega}(\varepsilon^{-1})$, $T = \tilde{\Omega}(d\varepsilon^{-1})$, $\eta = \tilde{\Theta}(d^{-1}\varepsilon)$, suppose $\theta_t$ is the angle between $\beta_t$ and $\mu$, then by running algorithm 1 with step size $\eta$ and batch size $B$, when $t < T - 1$, $\theta_t \geq \theta_{t+1}$ holds with probability at least $1 - \delta$, and with probability at least $1 - \delta$, $\theta_{T-1} \leq O(\varepsilon)$.*

Building on Lemma 3.1, assuming that pseudo-labels follow a Bernoulli distribution, and leveraging the relationship between the dot product of the classifier and the samples with $\theta_t$, we can readily derive the entropy variations for different samples and identify the regions to which different types of samples belong, as stated in the following theorem.

**Theorem 3.2.** *Under the assumptions of Lemma 3.1, let $d = 2$ and suppose $\hat{y}^{(t)}|x \sim \mathrm{Ber}(\vartheta(\beta_t^T x))$ is the pseudo-label of $x$, where $\vartheta(z) = \frac{1}{1+e^{-z}}$. Define $l(\alpha)$ as the line $x^T \alpha^\perp = 0$, where $\alpha^\perp$ is perpendicular to $\alpha$. Let $A(\alpha_1, \alpha_2)$ denote the region swept by $l(\alpha_1)$ rotating to $l(\alpha_2)$ along the trajectory of $\beta_{\mathrm{init}}$ towards $\mu$ during self-training. Denote $H_t(x) = \mathrm{Ent}[\hat{y}^{(t)}|x]$ be the entropy of $\hat{y}^{(t)}|x$. For $t < T - 1$, with probability at least $1 - \delta$, the entropy changes as follows: (i) $H_t(x)$ first decreases and then increases if $x \in A(\beta_0, \mu)$; (ii) $H_t(x)$ decreases if $x \in A(\mu, \beta_0^\perp)$; (iii) $H_t(x)$ first increases and then decreases if $x \in A(\beta_0^\perp, \mu^\perp)$; and (iv) $H_t(x)$ increases if $x \in A(\mu^\perp, \beta_0)$.*

Based on Theorem 3.2[1], entropy variation during the iterative process of self-training can be classified into four patterns: (i) a decrease followed by an increase; (ii) a consistent decrease; (iii) a decrease followed by an increase; and (iv) a consistent increase. These patterns are visualized in Figure 2(a), where the light blue, gray, dark blue, and yellow regions correspond to each respective entropy variation pattern. Specifically, we partition $\mathbb{R}^2$ using vectors $\beta_0, \mu, \beta_0^\perp, \mu^\perp$ and delineate the positions of the initial classifier $\beta_{\mathrm{init}}$ and optimal $\mu$. The updates to the classifier during self-training can be interpreted as the gradual rotation of the initial classifier $\beta_{\mathrm{init}}$ toward the Bayes-optimal classifier $\mu$. Moreover, due to the small initial error of $O(1)$ in the classifier, the angle $\theta_0$ between $\beta_0$ and $\mu$ is relatively small. As a result, the regions $A(\beta_0, \mu)$ and $A(\beta_0^\perp, \mu^\perp)$ occupy an relatively small proportion of $\mathbb{R}^2$, indicating that most samples exhibit monotonic entropy changes.

From experimental and theoretical analyses of self-training, we derive five key insights regarding entropy variation: (i) while the overall entropy of samples typically decreases, this trend does not hold for every individual sample; (ii) most samples exhibit monotonic entropy changes; (iii) the effectiveness of self-training arises from samples undergoing entropy reduction, which enables their transition from incorrect to correct predictions; (iv) in some cases, an increase in entropy could lead to a reversal of predictions, causing previously correct samples to become incorrectly classified; (v) these entropy variations are intrinsically linked to the spatial relationships between the samples and the classifiers $\beta_{\mathrm{init}}$ and $\mu$. These insights will help deepen our understanding of CoT from the perspective of entropy variation.

---

[1]Proofs of Lemma 3.1 and Theorem 3.2 are in Appendix A.1.

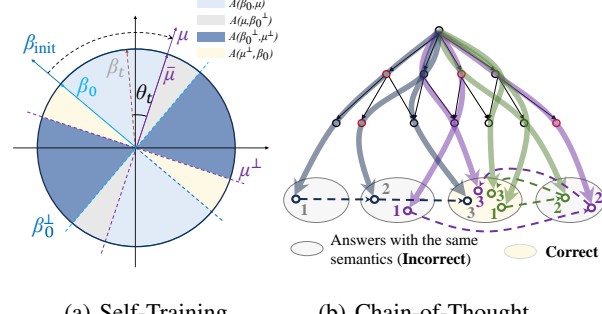

(a) Self-Training     (b) Chain-of-Thought

*Figure 2.* Visualizations of entropy variations in the iterative process of self-training and CoT reasoning. **In the self-training diagram**, the iterative process represents the gradual convergence of the initial classifier $\beta_{\text{init}}$ toward the Bayes optimal classifier $\mu$. At each iteration, changes in the angle between the classifier and the samples in different regions correspond to entropy variations within those samples. **In the CoT reasoning diagram**, each node in the directed acyclic graph represents a computation, with red nodes indicating erroneous computations. Each ellipse denotes a set of leaf nodes, corresponding to semantically equivalent answers, and the numbers indicate their respective iteration rounds. Bold arrows are used to represent complete reasoning paths. As the iterations proceed, these paths are gradually corrected. The semantic entropy at each iteration is determined by the distribution of generated answers across different semantic categories.

## 3.2. Semantic Entropy Variation in Chain-of-Thought

Chain-of-thought (CoT) reasoning, akin to self-training, relies on model-generated information to enhance task performance. Specifically, CoT aims to reduce prediction uncertainty in LLMs by leveraging intermediate reasoning processes generated by the models themselves. This uncertainty can be effectively quantified using semantic entropy (Farquhar et al., 2024). To formalize this concept, we begin by defining the LLM generation process and the mechanism of iterative CoT reasoning, then introduce the definition of semantic entropy, and finally discuss its parallels with self-training, drawing conclusions from this analogy.

Let $\textbf{LLM}(\cdot)$ denote a probabilistic language model. We write $s' \sim \textbf{LLM}(s; \tau)$ to indicate that the output $s'$ is sampled from the model given input $s$ and temperature parameter $\tau$. Based on this formulation, we next define how an LLM performs reasoning to solve complex problems.

**Definition 3.3** (**Reasoning Structures**). Given a question $q$, a prompt $p$, and a temperature parameter $\tau$, the set of reasoning–answer pairs $(r, a)$ generated by LLM is denoted by $\mathcal{J}(q, p; \tau)$, with its associated joint probability density denoted by $J$. A reasoning tree $\mathcal{T} = (\mathcal{V}, \mathcal{E})$ is a directed acyclic graph (DAG) constructed by the LLM to represent the computational steps from the question $q$ to potential answers. Each node $v \in \mathcal{V}$ is associated with a state $\mathfrak{s}(v) \in \{0, 1\}$ indicating whether the reasoning at that node is correct ($\mathfrak{s}(v) = 1$) or incorrect ($\mathfrak{s}(v) = 0$). A rea-

soning path $r$ is a specific path within $\mathcal{T}$ that begins at the root node corresponding to $q$ and terminates at a leaf node representing an answer $a$.

Building on the formalization of reasoning structures, we introduce the definition and computation of semantic entropy. For clarity, we present the formulation where the LLM input consists only of the question $q$ and the prompt $p$.

**Definition 3.4** (**Semantic Entropy**). Let $A = Answer(q, p; \tau)$ be the set of answers generated by an LLM for a given question $q$ and prompt $p$. Assume that this answer set can be partitioned into several disjoint semantic clusters $\mathcal{C} = \{C_i\}_{i \in [k]}$, such that $A = \bigcup_{i \in [k]} C_i$, where exactly one cluster contains the correct answer. Let $g(C \mid q, p; \tau)$ denote the probability distribution over $\mathcal{C}$. The semantic entropy is then defined as $e = \mathbb{E}_C[-\log g(C \mid q, p; \tau)]$. In practical use, let LLMs generate $N$ distinct answers $\hat{A} = \{a_i\}_{i \in [N]}$, which are then divided into $k$ semantic clusters $\hat{\mathcal{C}} = \{C_j\}_{j \in [k]}$. By normalizing, we obtain a discrete probability distribution $\{g_j\}_{j \in [k]}$, where $g_j = |C_j| / N, \sum_{j \in [k]} g_j = 1$. Consequently, we can use $\hat{e} = -\sum_{j=1}^{k} g_j \log g_j$ as an approximation of $e$.

In self-training, pseudo-labels guide the initial classifier toward the Bayes-optimal classifier. Similarly, in CoT, the reasoning process aims to steer the LLM's predictions towards the correct path, and thus, getting the correct answer. Building on this analogy and the discussion in Section 3.1, we propose the following definitions for CoT reasoning.

**Definition 3.5** (**Initial and Optimal Paths**). The initial reasoning path, $r_0$, is the reasoning path generated by LLM for a given question $q$ using an initial prompt $p_0$. The set of optimal reasoning paths, $R_{\text{opt}}$, comprises those reasoning paths $r \in R(q, p; \tau)$ for which the probability of deriving the correct answer, $a_{\text{correct}}$, $J_{a|r}(a_{\text{correct}}|r)$, is greater than or equal to a predefined high confidence threshold $\psi$.

In Section 3.1, the classifier $\beta$ is updated through information augmentation using unlabeled data. In contrast, the update of reasoning paths in CoT reasoning is achieved by in-depth problem analysis, transforming the problem into more tractable sub-problems, and awakening unused internal knowledge within LLM directly aiding problem-solving. Based on this, existing reasoning paths are updated through error correction or by considering different perspectives.

**Definition 3.6** (**Iterative CoT Reasoning**). Iterative CoT is a process that refines reasoning paths over successive iterations. A new reasoning path $(r_{t+1}, a_{t+1})$ is generated by LLM based on the original question $q$, an adaptive prompt $p'$, and the history of previously generated paths $r_0, \ldots, r_t$. The update from $r_t$ to $r_{t+1}$ typically involves identifying the first erroneous node $v_i$ in $r_t$ (where $\mathfrak{s}(v_i) = 0$), backtracking, and generating a corrected sub-path.

To further investigate how semantic entropy evolves within the CoT framework, we adopt the following assumption.

**Assumption 3.7.** As the semantic entropy $e$ decreases, the likelihood of the LLM with CoT reasoning producing a correct answer to question $q$ increases.

Building on the preceding definitions and Assumption 3.7, the iterative process of applying CoT reasoning to tackle complex questions can be interpreted as the LLM progressively producing and correcting to search for an optimal reasoning path within $R_{\text{opt}}$, starting from an initial reasoning path $r_0$. This process is often marked by a gradual reduction in semantic entropy, supported by empirical evidence presented in Figure 3. The decreasing entropy reflects the refinement of predictions, enabling the correction of initial errors and the generation of accurate answers.

Similar to self-training, not all questions show the expected reduction in semantic entropy during CoT reasoning. This phenomenon may arise from the presence of noisy information in reasoning processes, which could misdirect the reasoning trajectory of LLMs. To verify this, we conducted a simple experiment involving three rounds of iterative CoT reasoning on the AQuA arithmetic dataset, analyzing the semantic entropy variation of LLMs. The results, presented in Figure 8(b) in Appendix, reveal three distinct patterns of semantic entropy variation: (i) monotonic increase or decrease; (ii) increase followed by decrease or decrease followed by increase; (iii) no change. These patterns are visualized in Figure 2(b). For example, with a sampling number $N = 3$ for reasoning processes, if the first iteration yields three semantically distinct answers, this reflects high semantic uncertainty. If the second iteration converges to two distinct answers, uncertainty is reduced. By the third iteration, if all processes yield the same semantic answer, uncertainty may reduce to zero, indicating the LLM has identified an optimal reasoning path leading to the correct answer. Moreover, the other patterns of semantic entropy variation can also be similarly derived.

Although iterative CoT reasoning does not update model parameters, it dynamically adjusts the context (the set of reasoning paths). This process essentially achieves a tightening of the distribution in the generation space. That is, the longer a correct sub-path $r'$ within a reasoning path $r$ becomes, the smaller the scope of the answer set at the terminus of $r'$ (*i.e.,* the leaf nodes of the reasoning tree) (see Figure 2(b)). Consequently, the proportion of the cluster corresponding to the correct answer increases, leading to a decrease in semantic entropy. This reduction in semantic entropy, equivalent to an increased probability of the correct answer cluster, aligns with the mathematical objective of label distribution entropy reduction in self-training.

Based on the insights gained from self-training in Section 3.1 and the experimental analysis of CoT reasoning, we de-

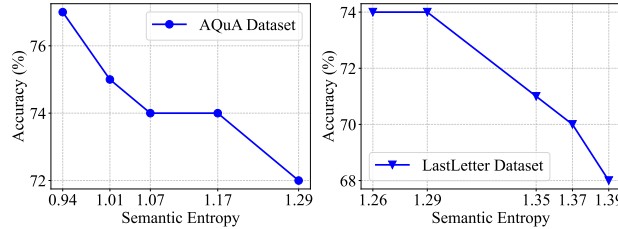

*Figure 3.* Accuracy under varying levels of semantic entropy on the AQuA and LastLetters datasets (based on 100 sampled instances).

rive five important conclusions regarding semantic entropy variations in CoT reasoning: (i) although the overall semantic entropy of questions generally decreases, this trend does not apply to every individual question; (ii) when the reasoning processes of the new iteration are excessively similar to those in the previous iteration, the semantic entropy of certain questions may exhibit no changes; (iii) the effectiveness of CoT reasoning stems from questions undergoing semantic entropy reduction, facilitating a transition from incorrect to correct predictions; (iv) for some questions, an increase in entropy could lead to prediction reversals, where previously correct predictions become incorrect; (v) these variations in semantic entropy are tied to the spatial relationships between the initial state and the optimal reasoning paths of the LLMs with CoT reasoning on a given question.

We will demonstrate in the next section how these findings can be utilized to further enhance CoT performance.

## 4. Methodology

Building on the conclusions from the analysis of semantic entropy variation in CoT reasoning presented in Section 3.2, we propose a novel CoT framework to improve reasoning performance. Specifically, we design a task-specific prompt module in Section 4.1 to guide the LLMs in generating high-quality initial reasoning processes. Next, in Section 4.2, we introduce an adaptive reasoning iteration module to refine the reasoning process and address the limitations of traditional CoT approaches, *i.e.,* over-reasoning and excessive similarity between consecutive reasoning iterations. An overview of our proposed CoT framework is provided in Figure 4. Detailed information about the task-specific prompt module is provided in Figure 6 in Appendix.

### 4.1. Task-Specific Prompt

The conclusion (v) presented in Section 3.2 underscores the pivotal role of CoT reasoning quality in the initial iteration. A high-quality initial CoT reasoning process effectively reduces the distance between the initial state and the optimal reasoning paths, thereby decreasing the number of iterations required for the LLMs to converge to the correct answers. Given that the initial CoT reasoning process is generated by

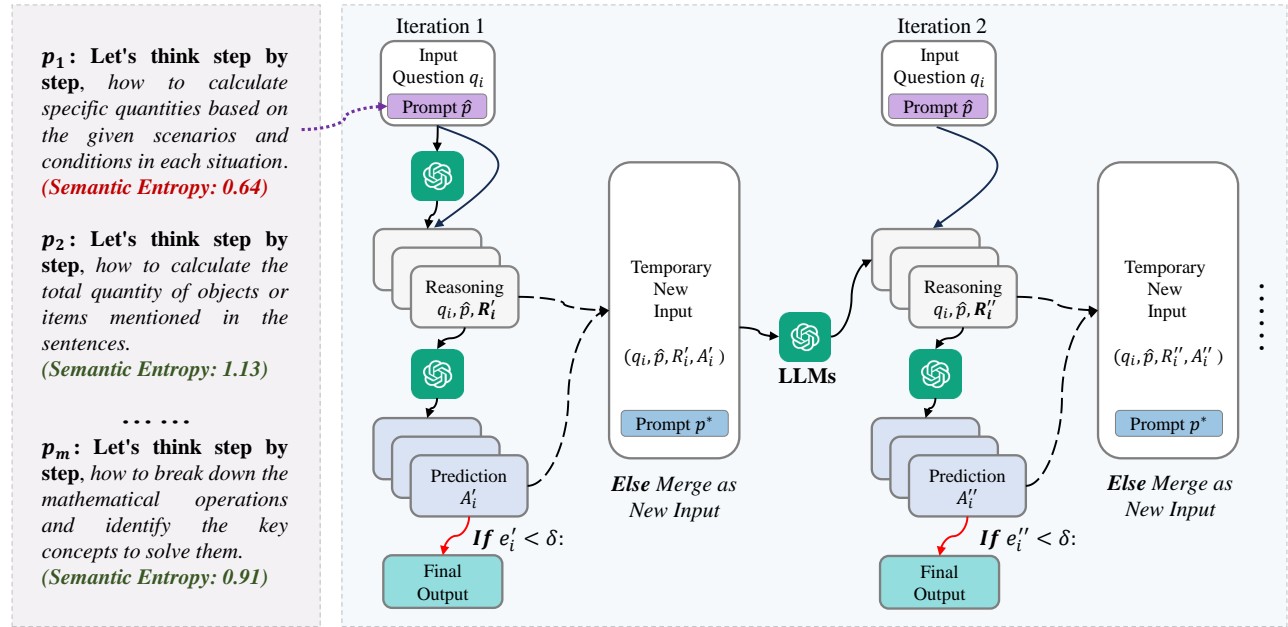

*Figure 4.* The flowchart of the proposed CoT framework consists of two key modules, *i.e.,* **Task-Specific Prompt** (light purple block) and **Adaptive Reasoning Iteration** (light blue block). Specifically, the task-specific prompt module first utilizes LLMs to generate $m$ candidate prompts and evaluates their semantic entropy on the given dataset. The prompt with the lowest entropy is selected as the optimal prompt $\hat{p}$, providing guidance for the subsequent adaptive reasoning iteration module to produce high-quality initial reasoning. In the adaptive reasoning iteration module, the uncertainty is calculated at each iteration and compared to a predefined threshold $\delta$. This evaluation determines whether to accept the current prediction as the final output or to proceed to another iteration. If the uncertainty remains high, a new reasoning round is initiated with a new prompt $p^*$, designed to introduce diversity compared to previous reasoning steps. This iterative process continues until the uncertainty is substantially reduced or the maximum number of iterations is reached.

the LLMs based on the provided prompts, this highlights the critical necessity of optimizing prompts to enhance the quality of reasoning processes from the outset.

Previous CoT methods (Wei et al., 2022; Ling et al., 2024) often rely on general prompts (*e.g.,* "Let's think step by step") to guide the reasoning process. Although such prompts are effective for generic tasks, they frequently fall short of capturing the nuances of domain-specific or fine-grained tasks, resulting in inadequate reasoning processes. To address this limitation, we propose a task-specific prompt module that automatically searches for the optimal prompt tailored to the task's characteristics. Specifically, our approach begins by constructing an instruction: **# Instruction:** *"Let's think step by step" is a general prompt that can guide the LLMs to produce reasoning processes. However, in specialized domains, this prompt may lack accuracy and clarity. Below is a dataset sample. Please enhance the "Let's think step by step, %s" prompt by adding a sentence in the %s section to better fit the dataset's characteristics.*

We then sample a question set $Q' = \{q'_1, q'_2, \cdots, q'_k\}$ from the dataset $\mathcal{Q}$, representing the task distribution. This set $Q'$ is concatenated with the instruction and fed into the LLMs, which performs $m$ rounds of sampling to generate a candidate task-specific prompt set $P = \{p_1, p_2, \cdots, p_m\}$.

Next, we sample another disjoint question set $Q'' = \{q''_1, q''_2, \cdots, q''_k\}$ from $\mathcal{Q}$, ensuring $Q' \cap Q'' = \emptyset$. Each candidate prompt from $P$ is concatenated with the questions in $Q''$ and used for zero-shot CoT inferences (Kojima et al., 2022). During inference, the mean semantic entropy for all questions in $Q''$ is calculated under each candidate prompt, yielding in the set $E = \{e_1, e_2, \cdots, e_m\}$. For clarity, we define a simple mapping function $f : P \rightarrow E$ to represent the relationship between candidate prompts and their corresponding mean semantic entropy values.

Based on conclusions (i) and (iii) from the analysis of semantic entropy variation in Section 3.2, we can infer that lower semantic entropy corresponds to better CoT performance. Consequently, the prompt with the minimal semantic entropy can be regarded as the optimal prompt:

$$\hat{p} = f^{-1}\left(e_{\arg\min_i\{e_i|e_i \in E\}}\right), \qquad (1)$$

where $f^{-1}(\cdot)$ function maps the minimum mean semantic entropy value back to its corresponding prompt $\hat{p}$. This process ensures that the selected prompt $\hat{p}$ minimizes the semantic uncertainty associated with the specific task, thereby improving the quality of initial reasoning processes.

During the testing phase, we concatenate the optimal prompt $\hat{p}$ derived from Eq. (1) with the question, and input the com-

bined text into the LLMs. Following the self-consistency CoT approach (Wang et al., 2022), we perform $N$ rounds of sampling to generate diverse reasoning processes:

$$R'_i = \{\mathbf{LLM}(\mathbf{Concat}(q_i, \hat{p}))_j \mid j = 1, 2, \ldots, N\}, \quad (2)$$

where $\mathbf{LLM}(\cdot)_j$ denotes the $j$-th sampling result produced by the LLMs, $R'_i = \{r'_{i,1}, r'_{i,2}, \ldots, r'_{i,N}\}$ represents the set of reasoning processes generated, $q_i$ corresponds to the $i$-th question in the dataset $\mathcal{Q}$, and the $\mathbf{Concat}(\cdot)$ function refers to the sequential concatenation of the specified texts. Next, each of the $N$ reasoning processes obtained from Eq. (2) is individually concatenated with the question $q_i$ and the prompt $\hat{p}$. The resulting concatenated texts are then fed into the LLMs to generate predictions:

$$A'_i = \{\mathbf{LLM}(\mathbf{Concat}(q_i, \hat{p}, r'_{i,j})) \mid j = 1, \ldots, N\}, \quad (3)$$

where $A'_i = \{a'_{i,1}, a'_{i,2}, \ldots, a'_{i,N}\}$ represents the set of predictions generated by LLMs. At this stage, two options are available: (i) select the most frequent class from the $N$ answers in $A'_i$ as the final output (iteration terminates); (ii) concatenate the question $q_i$, the reasoning processes $R'_i$, and the predictions $A'_i$ for a new round of reasoning and prediction. If option (ii) is chosen, when should the iteration be stopped? Meanwhile, how can we ensure that the newly generated reasoning and predictions surpass the previous ones? These issues will be discussed in depth in Section 4.2, where corresponding solutions will also be proposed.

## 4.2. Adaptive Reasoning Iteration

The conclusion (iv) presented in Section 3.2 highlights that once the reasoning processes guide the LLMs to predictions with low uncertainty, deeper iterations do not further reduce semantic entropy. Instead, such iterations often introduce noisy information, undermining predictive accuracy. This phenomenon, which we term over-reasoning, risks altering correct early predictions during subsequent iterations. To address this, we calculate the semantic entropy $e'_i$ of the predictions $A'_i$ and compare it against a predefined threshold $\delta$. If $e'_i \leq \delta$, the predictions $A'_i$, derived from reasoning processes $R'_i$, is accepted as the final output. By leveraging semantic entropy to quantify the uncertainty of LLMs, we can stop iterations at the right moment to output predictions, effectively mitigating the risk of over-reasoning.

Conversely, if $e'_i > \delta$, the reasoning process proceeds to the next iteration to further reduce uncertainty. A naive solution involves concatenating the reasoning processes $R'_i$, predictions $A'_i$, and question $q_i$ from the previous iteration while reusing the prompt $\hat{p}$. However, this approach often results in new reasoning processes that closely resemble previous iterations. As demonstrated by the conclusion (ii) presented in Section 3.2, such high similarity could hinder further entropy reduction. To overcome this, we propose introducing

greater divergence between reasoning iterations, enabling exploration of alternative paths and enhancing the ability of LLMs to solve complex questions. Specifically, we design a new prompt $p^*$ to replace the $\hat{p}$, fostering a departure from prior reasoning steps: **#** $p^*$: *Based on the above thoughts, reevaluate from alternative perspectives to produce deeper, solution-oriented insights that go beyond prior inferences. Focus on identifying unexplored assumptions or challenges in the question context, and propose new processes.*

Then, we concatenate the new prompt $p^*$, the original question $q_i$, the previous reasoning processes $R'_i$, and the previous predictions $A'_i$ to generate a new round of reasoning:

$$R''_i = \{\mathbf{LLM}(\mathbf{Concat}(q_i, \hat{p}, r'_{i,j}, a'_{i,j}, p^*)) \mid j = 1, \ldots, N\}, \quad (4)$$

the prompt $p^*$ encourages LLMs to critically reflect on prior information, producing reasoning steps $R''_i$ that surpass and differ from the previous $R'_i$. To ensure sufficient divergence between new and previous reasoning, we measure their similarity using the Jaccard index (Jadhao & Agrawal, 2016):

$$s''_i = \mathbf{Sim}(R''_i, R'_i) = \frac{|R''_i \cap R'_i|}{|R''_i \cup R'_i|}, s''_i \in \mathbb{R}, \quad (5)$$

if $s''_i$ is greater than a predefined threshold $\tau$, Eq. (4) is reapplied for resampling until the condition is met. Finally, the LLMs discard the earlier reasoning $R'_i$ and the prediction $A'_i$, generating new predictions based solely on $R''_i$:

$$A''_i = \{\mathbf{LLM}(\mathbf{Concat}(q_i, \hat{p}, r''_{i,j})) \mid j = 1, \ldots, N\}\}, \quad (6)$$

After obtaining $A''_i$ via Eq. (6), its semantic entropy $e''_i$ is computed and compared to $\delta$ again. If $e''_i > \delta$, the process repeats (Eqs. (4) - (6)) until the uncertainty drops to $\delta$ or the maximum iteration count $T$ is reached.

When semantic entropy remains above $\delta$ after $T$ iterations, three scenarios may explain this outcome: (i) previous iterations contained valid reasoning steps, but randomness in LLMs sampling introduced biases in calculating semantic entropy. Increasing $N$ may mitigate this issue; (ii) effective reasoning paths exist but have yet to be discovered by LLMs. Extending $T$ could allow LLMs to identify such paths; (iii) the limitations of LLMs make the question inherently unsolvable, rendering further iterations unproductive.

Although increasing $T$ can enhance performance in the second scenario, it also escalates time and computational costs. To strike a balance between performance and efficiency, the strategy proposed in this paper is to stop further iterations when the entropy remains above $\delta$ after $T$ iterations. Instead, we apply majority voting across the predictions from all iterations, selecting the most frequent prediction as the final output. This approach enhances overall reliability while effectively reducing computational overhead.

| Method | Arithmetic | | | | | | Commonsense | | Symbolic | | Overall |
|---|---|---|---|---|---|---|---|---|---|---|---|
| | MultiArith | GSM8K | SingleEq | AddSub | AQuA | SVAMP | STQA | CSQA | Letter | Coin | Avg. (%) |
| RE2 | 91.7 | 76.3 | 81.5 | 72.9 | 54.3 | 78.3 | 66.2 | 74.9 | 57.0 | 91.2 | 74.4 |
| RE2 + SC | 96.2 | 77.3 | 87.0 | 80.0 | 60.6 | 83.2 | **68.6** | 77.5 | 63.8 | **98.0** | 79.2 |
| Zero-Shot | 51.2 | 10.8 | 62.4 | 56.7 | 38.6 | 56.3 | 66.2 | 74.5 | 1.4 | 50.2 | 46.8 |
| Zero-Shot-CoT | 92.8 | 74.7 | 84.4 | 74.7 | 55.5 | 77.0 | 63.5 | 73.6 | 55.0 | 93.4 | 74.5 |
| Zero-Shot-CoT + SC | 95.7 | 79.2 | 88.8 | 81.3 | 63.0 | 82.2 | 65.9 | 75.3 | 66.2 | 97.2 | 79.5 |
| + TSP | 97.0 | 81.1 | 90.0 | 84.8 | 65.7 | 85.5 | 66.7 | 76.7 | 68.4 | 97.6 | 81.4 |
| | (+1.3) | (+1.8) | (+1.2) | (+3.5) | (+2.7) | (+3.3) | (+0.8) | (+1.3) | (+2.2) | (+0.4) | (+1.9) |
| + ARI | 96.7 | 82.6 | 92.1 | 87.1 | 69.3 | 87.1 | 67.5 | **77.5** | 75.8 | 97.2 | 83.3 |
| | (+1.0) | (+3.4) | (+3.3) | (+5.8) | (+6.3) | (+4.9) | (+1.6) | (+2.2) | (+9.6) | (+0.0) | (+3.8) |
| + TSP + ARI | **98.2** | **83.0** | **92.9** | **88.4** | **70.1** | **87.5** | 66.7 | 76.7 | **77.2** | 96.4 | **83.7** |
| | (+2.5) | (+3.8) | (+4.1) | (+7.1) | (+7.1) | (+5.3) | (+0.8) | (+1.4) | (+11.0) | (-0.8) | (+4.2) |

*Table 1.* Accuracy (%) across ten reasoning datasets from three categories of zero-shot reasoning tasks. The number of self-consistency (SC) sampling is fixed at 3 for all cases. Blue and red fonts indicate increases and decreases in task performance compared to the "Zero-Shot-CoT + SC" method, respectively, while bold font highlights the best performance in each column.

# 5. Experiments

## 5.1. Experimental Settings

We evaluate our CoT framework on 10 reasoning datasets, including six arithmetic datasets (*i.e.,* MultiArith (Roy & Roth, 2016), GSM8K (Cobbe et al., 2021), SingleEq (Koncel-Kedziorski et al., 2015), AddSub (Hosseini et al., 2014), AQuA (Ling et al., 2017), and SVAMP (Patel et al., 2021)), two commonsense reasoning datasets (*i.e.,* StrategyQA (Geva et al., 2021) and CommonsenseQA (Talmor et al., 2018)), and two symbolic reasoning datasets (*i.e.,* LastLetter and CoinFlip (Wei et al., 2022)). We utilize GPT-3.5-turbo-0125 as the foundation model for all experiments, given its accessibility and cost-effectiveness.

Our evaluation adopts a progressive comparative approach. Specifically, we first test zero-shot reasoning by directly inputting questions into the LLMs without prompts. Next, we employ zero-shot CoT (Kojima et al., 2022), utilizing general prompts with greedy decoding to generate answers. Finally, we implement zero-shot CoT with self-consistency (Wang et al., 2022), which enhances accuracy through multiple decoding attempts and a voting mechanism. Building on these baselines, we propose a novel CoT framework comprising two modules, *i.e.,* task-specific prompt (TSP) and adaptive reasoning iteration (ARI). The TSP module improves the initial CoT reasoning process by replacing generic prompts with task-specific ones, while the ARI module further enhances task performance through iterative refinement of the reasoning process. Moreover, we also compared two popular CoT methods, *i.e.,* RE2 (Xu et al., 2024) and Contrastive-CoT (Chia et al., 2023).

## 5.2. Main Results

We followed literature (Kojima et al., 2022) to construct zero-shot reasoning tasks across all 10 datasets, and performed few-shot reasoning tasks on the MultiArith and

| Method | MultiArith | GSM8K | Avg. (%) |
|---|---|---|---|
| Contrastive-CoT | 89.7 | 69.4 | 79.6 |
| Contrastive-CoT + SC | 93.0 | 71.9 | 82.5 |
| Few-Shot | 78.3 | 53.8 | 66.1 |
| Few-Shot-CoT | 94.3 | 69.1 | 81.7 |
| Few-Shot-CoT + SC | 97.2 | 73.7 | 85.5 |
| + ARI | **97.3** | **77.4** | **87.4** |
| | (+0.1) | (+3.7) | (+1.9) |

*Table 2.* Accuracy across the MultiArith and GSM8K datasets from the arithmetic category of few-shot reasoning tasks. The number of self-consistency (SC) samplings is fixed at 3 for all cases.

GSM8K datasets. The results of these experiments are presented in Table 1 and Table 2, respectively.

For the zero-shot task, the baseline zero-shot method achieves an overall average accuracy of 46.8%. Incorporating CoT reasoning (Zero-Shot-CoT) significantly enhances performance, raising the accuracy to 74.5%. Further improvement is observed with the integration of self-consistency (SC), which increases the average accuracy to 79.5%. Building on this foundation, the addition of task-specific prompt (TSP) and adaptive reasoning iteration (ARI) modules further elevates the average accuracy to 83.7%. This represents a 4.2% improvement over the SC approach, demonstrating the advantages of our method across various task categories. Furthermore, our method showed the most significant performance on arithmetic datasets. Specifically, compared to the SC approach, the average performance across six datasets increased from 81.7% to 86.7%. This improvement can be attributed to the fact that deep reasoning enables the LLMs to systematically identify solution pathways. In contrast, the performance gains on commonsense datasets are relatively weak, even approaching the level of the zero-shot method. This limitation may arise from the dependency of these tasks on the prior knowledge of the LLMs. If the relevant commonsense knowledge was not encountered during pre-training, deep reasoning alone is insufficient to address the question.

In the few-shot task, the SC approach achieves an average accuracy of 85.4%. The incorporation of the ARI module results in a significant performance improvement, raising the accuracy to 87.4%. However, the TSP module is not applicable to few-shot tasks, as the LLMs have already utilized the limited examples provided to generate optimal reasoning in the initial iteration, rendering the reconstruction of a new reasoning process through TSP unnecessary.

Based on the above analysis, it is evident that the proposed ARI module enhances performance in both zero-shot and few-shot tasks, demonstrating its potential as a plug-and-play solution applicable across all CoT methods. Similarly, the TSP module can be applied to zero-shot methods to improve the quality of reasoning in the initial iteration. Ablation experiments presented in the last four rows of Table 2 confirm that, compared to the ARI module alone, the TSP module contributes an average performance improvement of 0.4%, thereby validating its effectiveness.

### 5.3. Adaptive Versus Fixed Iterative Reasoning

To further verify the effectiveness of our proposed method, we compared its accuracy and time costs with the fixed iterative reasoning approach on the AQuA dataset.

#### 5.3.1. Comparison of Accuracy and Time Costs

In terms of accuracy (see Figure 5(a)), our method demonstrates a clear advantage. It achieves 70.8% at the second iteration and maintains stability, reaching 71.3% by the fifth iteration. In contrast, the fixed iteration method shows slower improvement, peaking at 67.3% and then dropping to 62.6% by the fifth iteration. Regarding time cost (see Figure 5(b)), both methods exhibit a linear growth trend, but our method is significantly more time-efficient. The time cost of our approach increases gradually from 1 hour and 5 minutes at the first iteration to 4 hours and 18 minutes at the fifth iteration, reflecting a moderate growth rate. In comparison, the fixed iteration method follows a steeper trajectory, with the time cost rising from 1 hour and 5 minutes to 6 hours and 29 minutes by the fifth iteration.

Overall, our adaptive iteration outperforms fixed iteration in both effectiveness and efficiency, achieving an optimal balance between the two as early as the second iteration, highlighting its practicality and strong performance.

#### 5.3.2. Why Our Method Works?

Our proposed method significantly surpasses fixed reasoning iteration by effectively addressing two critical challenges: (i) over-reasoning, and (ii) high similarity between consecutive reasoning iterations. To resolve the issue of over-reasoning, we incorporate a mechanism that quantifies the prediction uncertainty of the LLMs using semantic

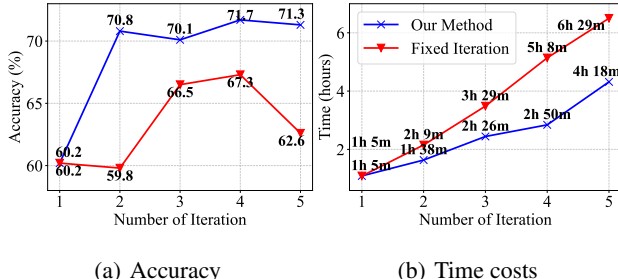

(a) Accuracy  (b) Time costs

*Figure 5.* Accuracy and time costs of adaptive reasoning iteration compared to the fixed reasoning iteration on the AQuA dataset.

entropy. The iteration process is terminated as soon as a low-uncertainty state is reached, ensuring predictions are made at the optimal stage. For example, as illustrated in Table 7, the semantic entropy after the first iteration falls below the predefined threshold of 0.95 (indicating that at least two out of three elements in the prediction set are consistent). At this point, further iterations are deemed unnecessary, and the current prediction is finalized as the output.

When the uncertainty is high and further iterations are required, the issue of high similarity between consecutive iterations may arise. To address this, we propose a novel prompt $p^*$ to encourage greater divergence in subsequent reasoning iterations, with the Jaccard index introduced to quantify this diversity. If insufficient diversity is detected, the reasoning process is resampled until predefined conditions are met. This strategy effectively ensures the diversity and independence of consecutive reasoning iterations. For example, as detailed in Table 7, the prompt $p^*$ and resampling strategy successfully guided the LLMs to generate more distinct reasoning paths, transforming an initial incorrect prediction into a correct one. Furthermore, as shown in Table 5, our method achieves lower similarity between consecutive iterations compared to existing approaches.

By dynamically adjusting reasoning iterations and promoting diversity in reasoning paths, our approach enhances accuracy while significantly reducing computational costs.

## 6. Conclusion

This paper explores the conceptual parallels between chain-of-thought (CoT) reasoning and self-training, identifying their shared objective of iteratively leveraging information augmentation to minimize prediction uncertainty. Through theoretical analysis and experimental validation, we reveal semantic entropy dynamics in CoT reasoning and propose improvements, including a task-specific prompt module to optimize initial reasoning processes and an adaptive reasoning iteration module to mitigate over-reasoning and enhance reasoning diversity in consecutive iterations. Collectively, these innovations significantly improve the performance of CoT reasoning in addressing complex tasks.

## Acknowledgements

This work was partially supported by the National Key Research and Development Program of China under Grant No. 2022YFA1004100, the Natural Science Foundation of Guangdong Province of China under Grant No. 2024A1515011381, and the National Natural Science Foundation of China under Grant No. 62476048.

## Impact Statement

This paper presents work whose goal is to advance the field of Machine Learning. There are many potential societal consequences of our work, none which we feel must be specifically highlighted here.

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

# A. Appendix

## A.1. Omitted Definitions and Proofs in Section 3.1

In this section, we supplement the definitions and detailed algorithm of self-training that were not elaborated in Section 3.1 and provide the proofs of Lemma 3.1 and Theorem 3.2. Our results primarily reference (Frei et al., 2022), and most definitions and algorithms will adopt the settings from their work. Here, we consider a very simple mixture Gaussian model (GMM) rather than adopting the more general but relatively more complex sub-exponential distribution with parameters $K, U, U', R$ as defined in (Frei et al., 2022).

**Definition A.1** (Gaussian Mixture Model). A joint distribution $(x, y) \sim \mathcal{D}$ over $\mathbb{R}^d \times \{\pm 1\}$ is called Gaussian mixture model, if $y \sim \text{Unif}(\{\pm 1\})$, and $x|y \sim \mathcal{N}(y\mu, I)$, where $\mu \in \mathbb{R}^d$ is the mean of $\mathcal{D}$. We use $\mathcal{D}_x$ to denote the marginal distribution of $\mathcal{D}$.

The information entropy of a discrete random variable $X$ is defined as $\text{Ent}[X] = \mathbb{E}[-\log X] = -\sum_{x \in \mathcal{X}} p(x) \log p(x)$, where $\mathcal{X}$ is the set of values that the random variable $X$ can take. A classifier for the Gaussian mixture model is given by $x \mapsto \text{sgn}(\beta^T x)$, where $\beta \in \mathbb{R}^d$ is an arbitrary vector. So we use $\beta$ to denote a classifier without saying $x \mapsto \text{sgn}(\beta^T x)$. According to Fact 3.4 in (Frei et al., 2022), the Bayes-optimal classifier of the Guassian mixture model defined in Definition A.1 is $\mu$. Assume we can access a initial classifier pseudo-labeler $\beta_{\text{init}}$, which is also called a pseudo-labeler, and the population error of $\beta_{\text{init}}$ is sufficiently small but constant. We then use a weight-normalized logistic regression method to train start from $\beta_{\text{init}}$ using only unlabeled samples. The loss function is $\ell(z) = \log(1 + \exp(-z))$. Let $\sigma > 0$ be temperature. The training dataset $S$ is partitioned into $T$ batches of size $B$. The general process of the self-training algorithm involves multiple iterations, where in each iteration, a batch of data is assigned pseudo-labels. The data labeled in this iteration, along with the existing labeled data and data pseudo-labeled in previous iterations, is used to update the model. This process continues until all samples have been assigned pseudo-labels. The detailed and formal algorithm description of self training using the pseudo-label strategy is presented in Algorithm 1.

---

**Algorithm 1** Self-Training

**Require:** Training dataset $S = \{x_i^{(t)}\}_{\substack{1 \leq i \leq B \\ 0 \leq t \leq T-1}}$, step size $\eta$, temperature $\sigma > 0$, initial pseudo-labeler $\beta_{\text{init}}$

1: $\beta_0 \leftarrow \beta_{\text{init}} / \|\beta_{\text{init}}\|$
2: **for** $t = 0, 1, \cdots, T-1$ **do**
3:      Generate pseudo-labels $\widehat{y}_i^{(t)} = \text{sgn}\left(\beta_t^T x_i^{(t)}\right)$ for batch $\{x_i^t\}_{1 \leq i \leq B}$
4:      $\tilde{\beta}_{t+1} = \beta_t - \frac{\eta}{B} \sum_{i=1}^{B} \nabla \ell\left(\frac{1}{\sigma} \cdot \widehat{y}_i^{(t)} \cdot \left(\beta_t^T x_i^{(t)}\right)\right)$
5:      $\beta_{t+1} = \tilde{\beta}_{t+1} / \|\tilde{\beta}_{t+1}\|$
6: **end for**
7: **return** $\beta_{T-1}$

---

### A.1.1. PROOF OF LEMMA 3.1

We first restate Lemma 3.1.

**Lemma A.2** (Lemma 3.1, restate). *Suppose $(x, y) \sim \mathcal{D}$ follows a mixture Gaussian models in $\mathbb{R}^d \times \{\pm 1\}$ with mean $\mu$ satisfying $\|\mu\| = \Theta(1)$, i.e., $y \sim \text{Unif}(\{\pm 1\})$ and $x|y \sim \mathcal{N}(y\mu, I)$. Let $\ell(z) = \log(1 + \exp(-z))$, and assume $\sigma \geq \max(1, \|\mu\|)$. Assume we can access a initial pseudo-labeler $\beta_{\text{init}}$ which satisfies $\Pr_{(x,y) \sim \mathcal{D}}[y \neq \text{sgn}(\beta_{\text{init}}^T x)] = O(1)$. Let $\varepsilon, \delta \in (0, 1)$, and assume that $B = \tilde{\Omega}(\varepsilon^{-1})$, $T = \tilde{\Omega}(d\varepsilon^{-1})$, $\eta = \tilde{\Theta}(d^{-1}\varepsilon)$, suppose $\theta_t$ is the angle between $\beta_t$ and $\mu$, then by running algorithm 1 with step size $\eta$ and batch size $B$, when $t < T - 1$, $\theta_t \geq \theta_{t+1}$ holds with probability at least $1 - \delta$, and with probability at least $1 - \delta$, $\theta_{T-1} \leq O(\varepsilon)$.*

*Proof.* To prove this lemma, we first present the results on sample complexity for labeled and unlabeled data obtained in (Frei et al., 2022) for self-training. Theorem 4.1 in (Frei et al., 2022) ensures that we can obtain a classifier with sufficiently small but constant error. More precisely, a standard logistic regression procedure produces a pseudolabler that can achieve the desired constant accuracy by using $O(d)$ labeled samples, which essentially represents the sample complexity of labeled data. As for the unlabeled data, the sample complexity is expressed in the lemma below.

**Lemma A.3** (The Sample Complexity for Unlabeled Data, Theorem 3.6 in (Frei et al., 2022)). *Suppose that $(x, y) \sim \mathcal{D}$ follows a mixture distribution with mean $\mu$ satisfying $\|\mu\| = \Theta(1)$ and parameters $K, U, U', R = \Theta(1)$. Let $\ell$ be well-behaved for some $C_\ell \geq 1$, and assume the temperature satisfies $\sigma \geq \max(R, \|\mu\|)$. Assume access to a pseudo-labeler $\beta_{\mathrm{init}}$ which satisfies $\mathrm{Pr}_{(x,y) \sim \mathcal{D}}(y \neq \mathrm{sgn}(\beta_{\mathrm{init}}^T x)) \leq C_{\mathrm{err}}$, where $C_{\mathrm{err}} = R^2/(72 C_\ell U')$. Let $\varepsilon, \delta \in (0, 1)$, and assume that $B = \tilde{\Omega}\left(\varepsilon^{-1}\right), T = \tilde{\Omega}\left(d\varepsilon^{-1}\right), \eta = \tilde{\Theta}\left(d^{-1}\varepsilon\right)$. Then with probability at least $1 - \delta$, by running Algorithm 1 with step size $\eta$ and batch size $B$, the last iterate satisfies $\mathrm{err}(\beta_{T-1}) \leq \mathrm{err}(\mu) + \varepsilon$. In particular, $T = \tilde{O}(d/\varepsilon)$ iterations using at most $TB = \tilde{O}(d/\varepsilon^2)$ unlabeled samples suffices to be within $\varepsilon$ error of the Bayes-optimal classifier.*

Some definitions mentioned in Lemma A.3 can be found in the original paper and are omitted here due to space constraints. We use the function $\ell(z) = \log(1 + \exp(-z))$ as our loss function and it's well behaved. For GMM, it is a mixture distribution with parameters $K, U, U', R = \Theta(1)$. Under the conditions of Lemma A.3, let $\overline{\mu} = \mu/\|\mu\|$. We can derive the following lemma regarding $\|\beta_t - \overline{\mu}\|$, which represents the distance between the classifier produced at each iteration of the algorithm and the normalized optimal classifier.

**Lemma A.4** (Recursion of $\Delta_t^2$, Lemma D.2 in (Frei et al., 2022)). *Suppose $\Delta_t^2 = \|\beta_t - \overline{\mu}\|^2$, then $\Delta_t^2$ satisfies that for $1 \leq t \leq T$,*

$$\Delta_t^2 \leq (1 - \eta/2C_g)\Delta_{t-1}^2 + \frac{\eta\varepsilon}{8C_g} + \frac{2C_d\eta^2}{\sigma^2},$$

*where $C_g, C_d, \sigma$ are all some positive constants such that $K = C_d C_g^2 \sigma^2 \geq 1$, $\Delta_0 \leq 2$.*

Note that there is close relationship between $\theta_t$ and $\Delta_t$, we can use the changes in $\Delta_t$ described in Lemma A.4 to characterize the changes in $\theta_t$, leading to the following lemma.

**Lemma A.5.** *Let $\theta_t$ denote the angle between $\beta_t$ and $\mu$, $\theta_t \in (0, \pi/2)$, and $\Delta_t^2 = \|\beta_t - \overline{\mu}\|^2$. Then for $1 \leq t < T - 1$, $\theta_t \geq \theta_{t+1}$, and $\theta_T \leq \varepsilon$.*

*Proof.* Since $\beta_t$ and $\overline{\mu}$ both have unit norm, it's easy to verify that

$$\|\beta_t - \overline{\mu}\|^2 = 2(1 - \cos\theta) = 4\sin^2\frac{\theta_t}{2}$$

It's sufficient to assume that $\theta_t \in (0, \pi/2)$ for any $t$, as the error rate of $\beta_{\mathrm{init}}$ is sufficiently small. Therefore $\Delta_t = 2\sin\frac{\theta_t}{2}$, i.e., $\theta_t = 2\arcsin\frac{\Delta_t}{2}$. This implies that $\theta_t$ and $\Delta_t$ share the same monotonicity, so it suffices to show that for $t < T$, $\Delta_t \leq \Delta_{t-1}$. By Lemma A.4, when $\eta = \frac{\varepsilon C_g}{16K}$ and $T \geq 32K\varepsilon^{-1}\log(32K\varepsilon^{-1})$, it's easy to verify that $\Delta_T \leq \varepsilon$, which means that $\Delta_t > \varepsilon$ for $t < T$. Hence, with the recursion of $\Delta_t^2$ in Lemma A.4, we have

$$\Delta_t^2 - \Delta_{t-1}^2 \leq -\frac{\eta}{2C_g}\Delta_{t-1} + \frac{\eta\varepsilon}{8C_g} + \frac{2C_d\eta^2}{\sigma^2}$$

$$= -\frac{\eta}{2C_g}\left(\Delta_{t-1} - \left(\frac{1}{4} + \frac{1}{4\sigma^4}\right)\varepsilon\right)$$

$$\leq -\frac{\eta}{2C_g}\left(\Delta_{t-1} - \varepsilon\right)$$

$$\leq 0$$

At the same time, $\theta_T = 2\arcsin\frac{\Delta_T}{2} = \Theta(\Delta_T) = O(\varepsilon)$. This concludes the proof. □

Finally, through the conditions and assumptions of Lemma A.3, along with the conclusion of Lemma A.5, we complete the proof of Lemma 3.1. □

### A.1.2. PROOF OF THEOREM 3.2

**Theorem A.6** (Theorem 3.2, restate)**.** *Under the assumptions of Lemma 3.1, let $d = 2$ and suppose $\hat{y}^{(t)}|x \sim \text{Ber}(\vartheta(\beta_t^T x))$ is the pseudo-label of $x$, where $\vartheta(z) = \frac{1}{1+e^{-z}}$. Define $l(\alpha)$ as the line $x^T \alpha^\perp = 0$, where $\alpha^\perp$ is perpendicular to $\alpha$. Let $A(\alpha_1, \alpha_2)$ denote the region swept by $l(\alpha_1)$ rotating to $l(\alpha_2)$ along the trajectory of $\beta_{\text{init}}$ towards $\mu$ during self-training. Denote $H_t(x) = \text{Ent}[\hat{y}^{(t)}|x]$ be the entropy of $\hat{y}^{(t)}|x$. For $t < T$, with probability at least $1 - \delta$, the entropy changes as follows: (i) $H_t(x)$ first decreases and then increases if $x \in A(\beta_0, \mu)$; (ii) $H_t(x)$ decreases if $x \in A(\mu, \beta_0^\perp)$; (iii) $H_t(x)$ first increases and then decreases if $x \in A(\beta_0^\perp, \mu^\perp)$; and (iv) $H_t(x)$ increases if $x \in A(\mu^\perp, \beta_0)$.*

*Proof.* Lemma 3.1 describes the trend in the changes of the vector corresponding to the classifier, which is particularly useful for analyzing entropy changes in $\mathbb{R}^2$, as the entropy of the pseudo-label distribution is directly related to the angle between the classifier and the samples. Since we define the pseudo-label distribution as a Bernoulli distribution using the sigmoid function, we first present the following lemma on the entropy of a Bernoulli distribution.

**Lemma A.7.** *Let $X \sim \text{Ber}(p), p \in [0, 1]$. Then $\text{Ent}[X]$ is a decreasing function of $|p - 1/2|$, i.e., if $X_1 \sim \text{Ber}(p_1)$, $X_2 \sim \text{Ber}(p_2)$, then $\text{Ent}[X_1] \geq \text{Ent}[X_2]$ if and only if $|p_1 - 1/2| \leq |p_2 - 1/2|$.*

*Proof.* For $X \sim \text{Ber}(p)$, $\text{Ent}[X] = f(p) = -p \log p - (1 - p) \log(1 - p)$. We define $0 \log 0 = 0$. It is easy to verify that $f(p)$ is symmetric about the line $x = 1/2$. On $(0, 1/2)$, $f(p)$ is monotonically increasing, and on $(1/2, 1)$, $f(p)$ is monotonically decreasing. Thus, for $X_1$ and $X_2$, $\max(p_1, 1 - p_1) \in (1/2, 1)$ and $\max(p_2, 1 - p_2) \in (1/2, 1)$, by the monotonicity and symmetry of $f(p)$, it is straightforward to conclude that $\text{Ent}(X_1) \geq \text{Ent}(X_2)$ if and only if $\max(p_1, 1 - p_1) \leq \max(p_2, 1 - p_2)$. Moreover, $\max(p_1, 1 - p_1) \leq \max(p_2, 1 - p_2)$ if and only if $|p_1 - 1/2| \leq |p_2 - 1/2|$, which completes the proof. $\square$

Now, we analyze the monotonic properties of $H_t(x)$. From Lemma A.7, we know that for $X \sim \text{Ber}(p)$, $\text{Ent}[X]$ depends on the magnitude of $|p - 1/2|$. In the subsequent proof, we establish the conclusion that during the iteration process, if $\vartheta(\beta_t^T x) < 1/2$, it will not happen that $\vartheta(\beta_{t+1}^T x) > 1/2$ unless $\vartheta(\beta_t^T x)$ is very close to $1/2$, causing $\vartheta(\beta_{t+1}^T x) > 1/2$ in the next iteration. This essentially states that in the self-training process, pseudo-labels do not fluctuate across iterations. Therefore, we only need to consider the relationship between $\vartheta(\beta_t^T x)$ and $\vartheta(\beta_{t+1}^T x)$.

Since $\vartheta(z) = \frac{1}{1+\exp(-z)}$ is monotonically increasing, it suffices to compare $\beta_t^T x$ and $\beta_{t+1}^T x$. Let the angle between $\beta_t$ and $x$ be $\theta^{(t)}(x) \in (0, \pi)$. Then, $\beta_t^T x = \|x\| \cos \theta^{(t)}(x) = \text{sgn}(\|x\| \cos \theta^{(t)}(x)) \|x\| |\cos \theta^{(t)}(x)|$. For the same sample, we only need to consider the changes in $\theta^{(t)}(x)$ as $t$ varies and the monotonicity of the function $f(z) = |\cos(z)|$ on $(0, \pi)$ ($f(z)$ is decreasing on $z \in (0, \pi/2)$ and increasing on $z \in (\pi/2, \pi)$). Define the angle between $x$ and $\mu$ as $\theta^{(\infty)}(x)$.

We now discuss the four regions defined in the theorem respectively.

1. $x \in A(\beta_0, \mu)$. When $\theta^{(0)}(x) \in (0, \pi/2)$ and $\theta^{(0)}(x) \leq \theta_0$, $x$ can be considered to lie between vector $\beta_0$ and vector $\mu$. If $\theta_t \geq \theta^{(t)}(x)$, then $\theta^{(t)}(x) = \theta_t - (\theta_0 - \theta^{(0)}(x)) \in (0, \pi/2)$; if $\theta_t \leq \theta^{(t)}(x)$, then $\theta^{(t)}(x) = (\theta_0 - \theta^{(0)}(x)) - \theta_t \in (0, \pi/2)$. Therefore, $\theta^{(t)}(x)$ first increases and then decreases, remaining within the interval $(0, \pi/2)$, which implies that $H_t(x)$ first decreases and then increases. The proof for the other case is similar.

2. $x \in A(\mu, \beta_0^\perp)$. When $\theta^{(0)}(x) \in (0, \pi/2)$ and $\theta^{(0)}(x) > \theta_0$, $x$ can be considered to lie between vector $\mu$ and $\beta_0^\perp$, with the angle between $\beta_0^\perp$ and $\mu$ lying in $(0, \pi/2)$. In this case, $\theta^{(t)}(x) = \theta_t + \theta^{(\infty)}(x) \in (0, \pi/2)$, so $\theta^{(t)}(x)$ decreases monotonically and remains in $(0, \pi/2)$, implying that $H_t(x)$ decreases monotonically. The other case is similar.

3. $x \in A(\beta_0^\perp, \mu^\perp)$. When $\theta^{(0)}(x) \in (\pi/2, \pi)$ and $\theta^{(\infty)}(x) \in (0, \pi/2)$, $x$ can be considered to lie between vector $\beta_0^\perp$ and $\mu^\perp$, where the angle between $\beta_0^\perp$ and $\mu$, as well as the angle between $\mu^\perp$ and $\beta_0^\perp$, both lie in $(0, \pi/2)$. In this case, $\theta^{(t)}(x) = \theta_t + \theta^{(\infty)}(x)$ decreases monotonically, but there exists some $t'$ such that $\beta_t$ becomes orthogonal to $x$, leading to $H_t(x)$ first increasing and then decreasing. The proof for the other case is similar.

4. $x \in A(\mu^\perp, \beta_0)$. When $\theta^{(0)}(x) \in (\pi/2, \pi)$ and $\theta^{(\infty)}(x) \in (\pi/2, \pi)$, $x$ can be considered to lie between $\mu^\perp$ and $-\beta_0$. In this case, $\theta^{(t)}(x) = \theta_t + \theta^{(\infty)}(x) \in (\pi/2, \pi)$ decreases monotonically, implying that $H_t(x)$ increases monotonically. The proof for the other case is similar.

Through the above analysis, we complete the proof of Theorem 3.2.

$\square$

### A.2. Rigorous Formalization of Reasoning Concepts Introduced in Section 3.2

The definitions that were not rigorously stated in Section 3.2 are presented here in detail.

**Definition A.8** (**LLM Generation Model**). Let $\Sigma$ be an alphabet, and $\Sigma^*$ be the set of all strings over $\Sigma$. For an input $s \in \Sigma^*$, and a temperature coefficient $\tau \in (0, 1)$, an LLM generation model is defined as a function $\mathbf{LLM} : \Sigma^* \times (0, 1) \to \mathcal{D}(\Sigma^*)$, where $\mathcal{D}(\Sigma^*)$ is the set of all probability distributions over $\Sigma^*$. We denote $s' \sim \mathbf{LLM}(s; \tau)$ as an output $s' \in \Sigma^*$ sampled from this LLM given the input $s$ and temperature $\tau$.

**Definition A.9** (**Reasoning-Answer Pairs and Sets**). Given a question $q$, a prompt $p$, and a temperature coefficient $\tau$: The set of all (Reasoning, Answer) pairs generated by the LLM under these conditions is denoted as

$$\mathcal{J}(q, p; \tau) = \{(r, a) : (r, a) \sim \mathbf{LLM}(q, p; \tau)\}$$

Its probability density function is denoted concisely as $J$. The set of reasoning paths generated by the LLM under these conditions is $R(q, p; \tau) = \{r : (r, a) \sim \mathbf{LLM}(q, p; \tau)\}$, with the probability density being the marginal distribution $J_r$ of $J$. The set of answers generated by the LLM under these conditions is $Answer(q, p; \tau) = \{a : (r, a) \sim \mathbf{LLM}(q, p; \tau)\}$, with the probability density being the marginal distribution $J_a$ of $J$. Furthermore, we define

$$Answer(q, p, r; \tau) = \{a' : (r', a') \sim \mathbf{LLM}(q, p, r; \tau)\}$$

as the set of all answers under a given reasoning path $r$, with its probability density being the conditional distribution $J_{a|r}$. The answer set $A$ can be partitioned into several semantic clusters: $A = \bigcup_{C \in \mathcal{C}} C$, where $\mathcal{C}$ is the set of these clusters, and exactly one cluster signifies the correct answer.

**Definition A.10** (**Initial Reasoning Path**). Given a question $q$, a zero-shot prompt $p_0$, and a temperature coefficient $\tau$, the initial reasoning path $r_0$ is defined such that $(r_0, a_0) \sim \mathbf{LLM}(q, p_0; \tau)$.

**Definition A.11** (Reasoning Tree Constructed by LLM, inspired by the definition in (Liu et al., 2024)). Given a question $q$, a prompt $p$, and a temperature coefficient $\tau$, the reasoning tree $\mathcal{T} = (\mathcal{V}, \mathcal{E})$ constructed by the LLM under these conditions is a directed acyclic graph, defined as follows:

- Each $v \in \mathcal{V}$ represents a computational node. Computational nodes generate content, including text, symbols, numerical values, or logical expressions. The computation at each node is defined as any combination of the following:

  1. **Symbols and text** (*e.g.,* variable definitions, natural language descriptions).
  2. **Computational expressions** (*e.g.,* arithmetic, algebraic operations).
  3. **Logical assertions** (*e.g.,* implication relations, hypothetical reasoning).
  4. **External knowledge references** (*e.g.,* theorems, formulas, common sense).

- Each $(u, v) \in \mathcal{E}$ represents a dependency relationship in reasoning, *i.e.,* the computation of $v$ depends on the content generated by $u$.

- The root node corresponds to $q$.

- $\mathcal{T}$ has multiple leaf nodes, some of which correspond to the correct answer $a_{\text{correct}}$.

- Each node $v \in \mathcal{V}$ has a state function $\mathfrak{s} : \mathcal{V} \to \{0, 1\}$, indicating whether the node is erroneous. Errors include, but are not limited to:

  1. Symbolic errors.
  2. Computational errors.
  3. Logical errors (*e.g.,* using external knowledge that does not meet requirements or is false, using incorrect logical proof methods).

**Definition A.12** (**Reasoning Path**). The reasoning path (or reasoning process) $r$ generated by an LLM for a question $q$, prompt $p$, and temperature coefficient $\tau$ is defined as a path $(q \to v_1 \to \cdots \to a)$ on the reasoning tree $\mathcal{T}$ constructed by the LLM under these conditions.

**Definition A.13** (**Optimal Reasoning Path**). The set of optimal reasoning paths $R_{\text{opt}}$ generated by an LLM for a question $q$, prompt $p$, and temperature coefficient $\tau$ is defined as $R_{\text{opt}} = \{r \in R(q, p; \tau) \mid J_{a|r}(a_{\text{correct}}|r) \geq \theta\}$, where $\psi$ is a confidence threshold, close to 1.

*Figure 6.* The proposed **Task-Specific Prompt** module tailors the prompt to the characteristics of a given task. Initially, a set of candidate prompts $\{p_1, p_2, \cdots, p_m\}$ is generated by incorporating a tailored instruction and a question set, $Q'$ sampled from the dataset. Next, another disjoint question set $Q''$ is sampled, distinct from $Q'$. The candidate prompts $\{p_1, p_2, \cdots, p_m\}$ are then evaluated based on their semantic entropy values $\{e_1, e_2, \cdots, e_m\}$, and the prompt $\hat{p}$ with the lowest entropy is selected as the optimal one for the task.

Due to the LLM's knowledge gaps and generation limitations, "optimal" needs to be defined as the most reliable path within the model's capabilities.

**Definition A.14 (Iterative CoT Reasoning).** Let $r_0$ be defined as in Definition A.10. Then, in an iterative CoT process, $(r_{t+1}, a_{t+1}) \sim \textbf{LLM}(q, p', r_0, \ldots, r_t; \tau)$, where $p'$ may be an adaptive prompt.

**Definition A.15 (Update of Reasoning Path in Iterative CoT).** For the current reasoning path $r_t$, a new path $r_{t+1}$ is generated as follows: the LLM identifies the node $v_i$ in $r_t$ closest to the root that satisfies $\mathfrak{s}(v_i) = 0$ (*i.e.,* is erroneous), reverts to node $v_{i-1}$, corrects the error, and generates a new sub-path, thereby forming $r_{t+1}$.

The conceptual parallels between self-training and chain-of-thought are summarized in Table 3.

| Self-Training | Chain-of-Thought |
|---|---|
| Distribution $\mathcal{D}$ over feature-label space $\mathcal{X} \times \mathcal{Y}$ | LLM's generation distribution $J$ of (Reasoning, Answer) pairs for a given $(q, p, \tau)$ |
| Classifier $\beta$ mapping an input $x \in \mathcal{X}$ to a label $y \in \mathcal{Y}$ | LLM generating an answer $a \in Answer(q, p, r; \tau)$ conditioned on a reasoning path $r$ |
| Initial classifier $\beta_0$ | Initial reasoning path $r_0$ generated from $(q, p_0, \tau)$ |
| Bayes-optimal classifier $\mu$ | Set of optimal reasoning paths $R_{\text{opt}}$ that yield $a_{\text{correct}}$ with high probability |
| Iterative classifier update: $\beta_t \rightarrow \beta_{t+1}$ | Iterative reasoning path update: $r_t \rightarrow r_{t+1}$ through error correction and refinement |
| Information augmentation for updates (*e.g.,* pseudo-labels from unlabeled data) | Information augmentation for updates (*e.g.,* deeper analysis of $q$, problem decomposition, prompting strategies to elicit new reasoning steps from LLM's internal knowledge) |

*Table 3.* Analogy between self-training and chain-of-thought reasoning.

## A.3. Task-Specific Prompt and Experimental Dataset Details

The specific details of the task-specific prompt module elaborated in Figure 6, and the particulars of the experimental datasets described in Table 4.

| Dataset | Answer Format (*1) | # of samples | Avg # words (*2) | Data split (filename) | License |
|---|---|---|---|---|---|
| SingleEq | N | 508 | 27.4 | questions.json | No License |
| AddSub | N | 395 | 31.5 | AddSub.json | Unspecified |
| MultiArith | N | 600 | 31.8 | MultiArith.json | Unspecified |
| GSM8K | N | 1319 | 46.9 | test.jsonl | MIT License |
| AQUA | M | 254 | 51.9 | test.jsonl | Apache-2.0 |
| SVAMP | N | 1000 | 31.8 | SVAMP.json | MIT License |
| CommonsenseQA | M | 1221 | 27.8 | dev_rand_split.jsonl | Unspecified |
| StrategyQA | Y | 2290 | 9.6 | task.json | Apache-2.0 |
| LastLetters | F | 500 | 15.0 | - | - |
| CoinFlip | Y | 500 | 37.0 | - | - |

*Table 4.* Detailed description of the datasets used in our experiments, highlighting their diversity and structure. (1) The "Answer Format" column indicates the type of responses expected for each dataset: N represents a numerical answer, M corresponds to selecting one option from multiple choices, Y indicates a binary answer (Yes or No), and F stands for free-form answers. (2) The "Avg # words" column represents the average number of words in the question texts, providing an estimate of their complexity.

### A.4. Parameter Sensitivity Analysis

Our proposed method involves two important hyper-parameters, *i.e.,* the number of iterations $T$, and the number of sampled reasoning paths $N$. We investigate the sensitivity of our method to these hyper-parameters on the AQuA dataset and report the results in Figure 5 and Figure 7. First, we fixed $N = 3$ and varied $T$ across the range of $\{1, 2, \cdots, 5\}$. The results indicate that our method is highly sensitive to the value of $T$. Accuracy increased sharply from 60.2% at the first iteration to 70.8% at the second iteration, after which the improvement plateaued. By the fourth iteration, accuracy reached 71.7%, with negligible changes in subsequent iterations. This trend suggests that with three reasoning paths, four iterations are sufficient to explore nearly all plausible solutions. Additional iterations yielded diminishing returns, likely constrained by the inherent reasoning capabilities of the employed LLMs. Moreover, the linear growth in computation time with increasing $T$ highlights the practical need to limit iterations for efficiency.

Next, we set $T = 3$ and varied $N$ from $\{1, 2, \cdots, 7\}$. The method exhibited significant sensitivity to $N$, with accuracy rising from 55.5% when using a single path to 70.9% with two paths. This underscores the method's capacity to aggregate diverse reasoning paths effectively. Additionally, increasing $N$ enhanced the accuracy of semantic entropy calculations, allowing more precise identification of samples prone to over-reasoning. However, this improvement came at the cost of steeply rising computation times, indicating a trade-off between accuracy and efficiency as $N$ increases.

### A.5. Additional Experimental Results

| Prompt Type | Iterations 1 & 2 | Iterations 2 & 3 | Avg. |
|---|---|---|---|
| General Prompt | 0.44 | 0.32 | 0.38 |
| Our Prompt $p^*$ | 0.28 | 0.29 | 0.29 |
| $\Delta$ | -0.16 | -0.03 | -0.10 |
| $\Delta\%$ | 36.4% | -9.4% | -26.3% |

*Table 5.* The reasoning similarity between new and previous iterations guided by general prompt and our $p^*$ on the AQuA dataset.

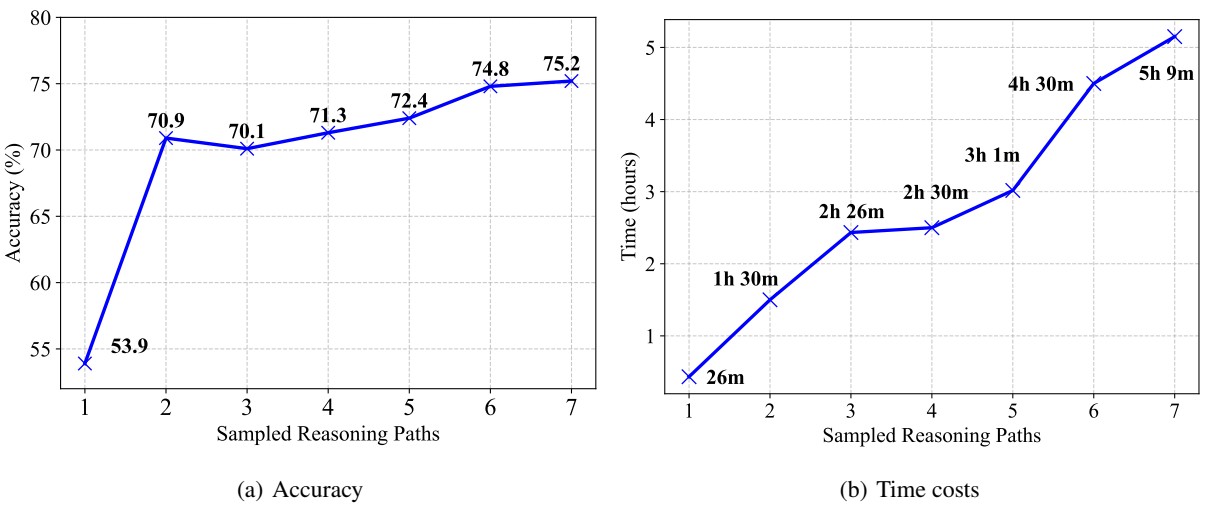

(a) Accuracy

(b) Time costs

*Figure 7.* Impact of the number of sampled reasoning paths on accuracy and time costs in our proposed method.

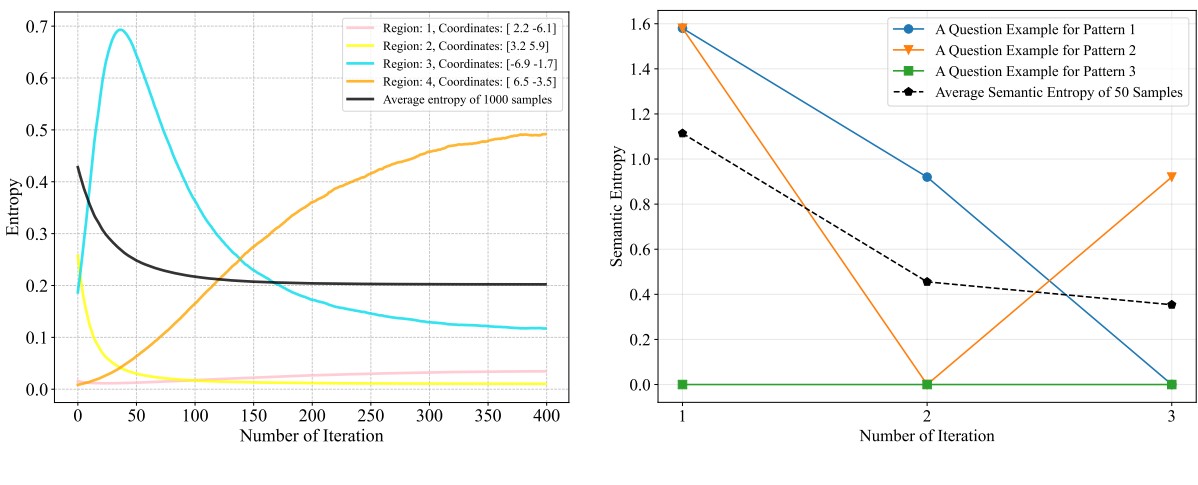

(a) Information entropy variation in self-training

(b) Semantic entropy variation in CoT reasoning

*Figure 8.* **Subfigure (a)**: entropy variation over iterations of self-training for four test samples in different regions. Sample 1 is in region $A(\beta_0, \mu)$, and its entropy first decreases and then increases. Sample 2 is in region $A(\mu, \beta_0^\perp)$, and its entropy decreases stably. Sample 3 is in region $A(\beta_0^\perp, \mu^\perp)$, and its entropy first increases then decreases steadily. Sample 4 is in region $A(\mu^\perp, \beta_0)$, and its entropy increases stably. During the iteration, the average entropy decreases until it stabilizes. **Subfigure (b)**: semantic entropy variation during iterations of CoT reasoning for questions across three patterns. In pattern 1, the semantic entropy of the question decreases gradually. In pattern 2, the semantic entropy remains constant. In pattern 3, the semantic entropy initially decreases before increasing.

| Question | Iteration 1 | Iteration 2 | Output |
|---|---|---|---|
| *Suppose you can travel from a place M to a place N by 3 buses, from place N to place O by 4 buses, from place O to place P by 1 buses and from place P to place Q by 3 buses. In how many ways can you travel from M to Q?* | **# Path 1:** We multiply the number of ways for each segment: from M to N (3 ways), N to O (4 ways), O to P (1 way), and P to Q (3 ways). This gives us: 3 * 4 * 1 * 3 = 36 ways. **# Path 2:** We have 3 options from M to N, 4 options from N to O, 1 option from O to P, and 3 options from P to Q. The answer is 3 * 4 * 1 * 3 = 36 ways. **# Path 3:** We calculate the number of ways for each segment: 27 ways from M to N, 256 ways from N to O, 1 way from O to P, and 27 ways from P to Q. Multiplying these together gives 27 * 256 * 1 * 27 = 186,624 ways. | **# Path 1:** I hope that explanation was helpful to you. If you have any more questions or need further assistance, feel free to ask! The answer is 186,624. **# Path 2:** That's correct! The total number of ways to travel from M to Q is 3 options for M to N multiplied by 4 options for N to O multiplied by 1 option for O to P multiplied by 3 options for P to Q, which equals 36 ways. **# Path 3:** To travel from M to Q, there are 27 ways from M to N, 256 ways from N to O, 1 way from O to P, and 27 ways from P to Q. Therefore, the answer is 186,624 ways. | 186,624 (Answer: 36) |
| *A starts travel towards south 3km, then travel 5 km towards east, and again travels 3 km to north, and finally travels 2km towards west. In the end how far is A from home?* | **# Path 1:** A starts at home, travels 3 km south, 5 km east, 3 km north, and 2 km west. After these movements, A is about 5.39 km from home, rounding to 5 km. **# Path 2:** A travels 3 km south, 5 km east, 3 km north, and then 2 km west. The movements north and south cancel each other out, as do the movements east and west, leaving A 2 km east of the starting point. **# Path 3:** A travels 3 km south, 5 km east, 3 km north, and 2 km west. The north and south distances cancel out, leaving A 3 km east, and the east-west difference results in a final position 5 km east of the starting point. | **# Path 1:** A travels south 3 km, east 5 km, north 3 km (canceling the southward distance), and west 2 km. Using the Pythagorean theorem, the final distance from home is 5.39km. The closest answer is 5 km. **# Path 2:** A travels 3 km south, 5 km east, 3 km north (canceling the southward distance), and 2 km west. This leaves A 2 km east of the starting point. **# Path 3:** Great job analyzing A's journey step by step! By considering how opposite movements cancel out, we can accurately determine the final position. Therefore, the answer is 4 km. | 5 km (Answer: 3 km) |

*Table 6.* **Over-reasoning** (row 2) and **high similarity between consecutive reasoning iterations** (row 3). In Iteration 2, Paths 1, 2, & 3 are updated based on their paths in Iteration 1. Over-reasoning occurs when the LLMs can already provide correct answers to the question in Iteration 1 but continue reasoning unnecessarily, introducing noise that leads to incorrect answers in Iteration 2. High similarity refers to cases where the corresponding paths in the two iterations are overly similar, failing to correct errors from the previous iteration.

| Question | Iteration 1 | Iteration 2 | Output |
|---|---|---|---|
| *Suppose you can travel from a place M to a place N by 3 buses, from place N to place O by 4 buses, from place O to place P by 1 buses and from place P to place Q by 3 buses. In how many ways can you travel from M to Q?* | **# Path 1:** We multiply the number of ways for each segment: from M to N (3 ways), N to O (4 ways), O to P (1 way), and P to Q (3 ways). This gives us: 3 * 4 * 1 * 3 = 36 ways. **# Path 2:** We have 3 options from M to N, 4 options from N to O, 1 option from O to P, and 3 options from P to Q. The answer is 3 * 4 * 1 * 3 = 36 ways. **# Path 3:** We calculate the number of ways for each segment: 27 ways from M to N, 256 ways from N to O, 1 way from O to P, and 27 ways from P to Q. Multiplying these together gives 27 * 256 * 1 * 27 = 186,624 ways. | *This iteration is skipped because the semantic entropy of the previous iteration is below the predefined threshold.* | 36 (Answer: 36) |
| *A starts travel towards south 3km, then travel 5 km towards east, and again travels 3 km to north, and finally travels 2km towards west. In the end how far is A from home?* | **# Path 1:** A starts at home, travels 3 km south, 5 km east, 3 km north, and 2 km west. After these movements, A is about 5.39 km from home, rounding to 5 km. **# Path 2:** A travels 3 km south, 5 km east, 3 km north, and then 2 km west. The movements north and south cancel each other out, as do the movements east and west, leaving A 2 km east of the starting point. **# Path 3:** A travels 3 km south, 5 km east, 3 km north, and 2 km west. The north and south distances cancel out, leaving A 3 km east, and the east-west difference results in a final position 5 km east of the starting point. | **# Path 1:** We reconsider A's movements as vectors: 3 km south, 5 km east, 3 km north, and 2 km west. The north and south movements cancel each other out, leaving a net displacement of 3 km east. **# Path 2:** A's total displacement from home can be calculated using the Pythagorean theorem by summarizing the journey. The net displacement is (5 - 2) km east and (3 - 3) km north, resulting in (3, 0). **# Path 3:** Excellent work breaking down A's journey step by step! By accounting for the cancellation of opposite movements, we can precisely calculate the final position. As a result, the answer is 4 km. | 3 km (Answer: 3 km) |

*Table 7.* Examples of how our method addresses the issues of over-reasoning and high similarity between consecutive reasoning iterations.

| Question | Iteration 1 | Iteration 2 | Output |
|---|---|---|---|
| *John likes to have lightly flavored tea every evening. In a 50% strong milk tea, he replaces 15% of it with milk twice. Then, he replaces 10 percent of the resultant solution with more milk. What is the final concentration of tea John drinks? Answer Choices: (A) 15.38% (B) 42% (C) 39.86% (D) 22.35% (E) 32.51%* | **# Path 1:** First, let's start with the 50% strong milk tea. John replaces 15% of it with milk twice. After the first replacement, the tea concentration becomes 50% * (1 - 15%) = 42.5%, and the milk concentration becomes 50% * 15% = 7.5%. After the second replacement, the tea concentration becomes 42.5% * (1 - 15%) = 36.125%, and the milk concentration becomes 7.5% + 42.5% * 15% = 14%. Next, John replaces 10% of this solution with more milk. The tea concentration becomes 36.125% * (1 - 10%) = 32.5125%, and the milk concentration becomes 14% + 36.125% * 10% = 17.6125%. Therefore, the final concentration of tea John drinks is approximately 32.51%. **# Path 2:** John starts with 50% strong milk tea and replaces 15% of it with milk twice. After the first replacement, 7.5% of tea is replaced, resulting in 42.5% tea. After the second replacement, 6.375% of tea is replaced, resulting in 36.125% tea. Then, John replaces 10% of this solution with more milk, which removes 3.6125% of tea, leaving 32.5125% tea. Therefore, the final concentration of tea is 32.51%. **# Path 3:** John starts with 50% strong milk tea and replaces 15% of it with milk twice. After the first replacement, 85% of the original tea remains, resulting in 42.5% tea. After the second replacement, 85% of the new mixture remains, resulting in 36.125% tea. The final concentration of tea is 36.125%, which is closest to answer choice (C) 39.86%. | *This iteration is skipped because the semantic entropy of the previous iteration is below the predefined threshold.* | (E) 32.51%
(Answer: (E)) |
| *Alex and Jacob works at a toy shop that make toys. Alex takes 7 hours to make a toy, and Jacob takes 9 hours to make a toy. During a month, both of them makes 35 toys in total. If both of them have worked for almost similar number of hours how many toys have been prepared by Jacob? Choices: (A) 15 (B) 16 (C) 17 (D) 18 (E) 19* | **# Path 1:** Alex takes 7 hours to make a toy, while Jacob takes 9 hours to make a toy. Together, they made a total of 35 toys, and both worked for almost the same number of hours. To determine how many toys Jacob made, we can calculate their individual production rates: Alex makes $\frac{1}{7}$ of a toy per hour, and Jacob makes $\frac{1}{9}$ of a toy per hour. Let $x$ represent the number of toys Jacob made. Since Alex made $35 - x$ toys, the total hours Alex worked would be $7(35 - x)$, and the total hours Jacob worked would be $9x$. Given that their working hours are nearly equal, we set up the equation $7(35 - x) = 9x$. Solving this, we get $245 - 7x = 9x$, which simplifies to $245 = 16x$, yielding $x = 15.3125$. Since the number of toys must be a whole number, we round down to conclude that Jacob made **15 toys**. **# Path 2:** To determine how many toys Jacob made, we first calculate the combined work rate of Alex and Jacob. Alex makes $\frac{1}{7}$ of a toy per hour, and Jacob makes $\frac{1}{9}$ of a toy per hour, giving a combined rate of $\frac{1}{7} + \frac{1}{9} = \frac{16}{63}$ toys per hour. Since both Alex and Jacob worked for approximately the same number of hours, let $x$ represent the hours each worked. The total hours worked by both is $2x$, and the total number of toys made is 35. Using the combined work rate, we set up the equation $\frac{16}{63} \times 2x = 35$. Solving for $x$, we get $\frac{32x}{63} = 35$, which simplifies to $32x = 2205$ and $x \approx 68.90625$ hours. To find the number of toys Jacob made, we multiply his work rate by the hours he worked: $\frac{1}{9} \times 68.90625 \approx 7.65625$. Since the number of toys must be a whole number, Jacob made approximately **8 toys**. However, this answer does not match any of the provided choices, indicating a discrepancy in the problem setup or calculations. **# Path 3:** Alex takes 7 hours to make a toy, so in one hour, he makes 1/7 of a toy, while Jacob takes 9 hours to make a toy, making 1/9 of a toy in one hour. Let A and J represent the number of toys Alex and Jacob made, respectively. Together, they made 35 toys in a month, which gives the equation: A/7 + J/9 = 35. Since they worked almost the same number of hours, the ratio of toys they made should be close to the ratio of their working speeds, which is 9:7, since Alex takes 7 hours to make a toy and Jacob takes 9 hours. Therefore, the number of toys Alex made should be 9x, and the number of toys Jacob made should be 7x. Given that the total number of toys is 35, we have 9x + 7x = 35, which simplifies to 16x = 35. Solving for x gives x = 35/16 = 2.1875. Since the number of toys must be whole, we round to the nearest whole numbers: Alex made 9 * 2 = 18 toys, and Jacob made 7 * 2 = 14 toys. Therefore, the answer is 18 toys. Hence, the correct answer is (D) 18. | **# Path 1:** I agree with your approach. Let's solve the equation step by step to find the number of toys made by Jacob. Let x represent the number of toys made by Jacob, the number of toys made by Alex is 35 - x. Given that Alex takes 7 hours to make a toy and Jacob takes 9 hours, the total number of hours spent by both making toys should be equal, leading to the equation 7 * (35 - x) = 9 * x. Expanding the left side, we get 245 - 7x = 9x. Rearranging the equation gives 245 = 16x. Dividing both sides by 16, we find x = 245 / 16 = 15.3125. Since x represents the number of toys made by Jacob and it cannot be fractional, we round down to the nearest whole number, so Jacob made 15 toys. Therefore, the correct answer is (A) 15. **# Path 2:** Let's consider the total amount of work done by Alex and Jacob in terms of the number of toys they each made. If Alex takes 7 hours to make a toy, then in x hours, he would make x/7 toys. If Jacob takes 9 hours to make a toy, then in x hours, he would make x/9 toys. Given that both of them together made 35 toys in total, we can set up the equation: x/7 + x/9 = 35. To solve for x, we need to find a common denominator, which in this case would be 63, so the equation becomes (9x + 7x) / 63 = 35. Simplifying this gives 16x / 63 = 35, which leads to 16x = 35 * 63, or 16x = 2205. Solving for x gives x = 2205 / 16 = 137.8 hours. Since we are looking for the number of toys made by Jacob, we substitute x back into Jacob's rate, so the number of toys made by Jacob is x/9 = 137.8 / 9 = 15.3. Therefore, Jacob made approximately 15 toys, and since we are looking for the nearest whole number, Jacob made 15 toys. **# Path 3:** Alex takes 7 hours to make a toy, so in one hour, he makes 1/7 of a toy. Jacob takes 9 hours to make a toy, so in one hour, he makes 1/9 of a toy. Together, they made 35 toys, so we have the equation: A/7 + J/9 = 35. Since their working speeds are in the ratio of 9:7, we assume Alex made 9x toys and Jacob made 7x toys. Solving for x, we get x = 35/16 = 2.1875. Rounding to the nearest whole number, Alex made 18 toys and Jacob made 14 toys. Therefore, the answer is 18 toys (D). | (A) 15
(Answer: (A)) |

*Table 8.* Examples of how our method addresses the issues of over-reasoning and high similarity between consecutive reasoning iterations.

| Question | Iteration 1 | Iteration 2 | Output |
|---|---|---|---|
| *The ratio of the volumes of a cube to that of the sphere which will fit inside the cube is? Answer Choices: (A) 2: π (B) 7:2 (C) 8:2 (D) 6: π (E) 8:3* | **# Path 1:** The volume of a cube is given by $V_{\text{cube}} = s^3$, where $s$ is the side length, while the volume of a sphere is calculated as $V_{\text{sphere}} = \frac{4}{3}\pi r^3$, with $r$ being the radius. When a sphere fits perfectly inside a cube, the sphere's diameter equals the cube's side length, making the radius $r = \frac{1}{2}s$. Substituting this into the sphere's volume formula yields $V_{\text{sphere}} = \frac{1}{6}\pi s^3$. The ratio of the cube's volume to the sphere's volume is therefore $V_{\text{cube}} : V_{\text{sphere}} = s^3 : \frac{1}{6}\pi s^3 = 6 : \pi$. Thus, the correct ratio is $6 : \pi$, corresponding to option (D). **# Path 2:** First, consider the volume of a cube and a sphere. The volume of a cube is $V_{\text{cube}} = s^3$, where $s$ is the side length, and the volume of a sphere is $V_{\text{sphere}} = \frac{4}{3}\pi r^3$, where $r$ is the radius. If the sphere fits perfectly inside the cube, its diameter equals the side length of the cube, so $2r = s$, which gives $r = \frac{s}{2}$. Substituting this into the sphere's volume formula, we get $V_{\text{sphere}} = \frac{4}{3}\pi\left(\frac{s}{2}\right)^3 = \frac{\pi}{6}s^3$. Comparing the volumes, the ratio of the cube's volume to the sphere's volume is $V_{\text{cube}} : V_{\text{sphere}} = s^3 : \frac{\pi}{6}s^3 = 6 : \pi$. Thus, the ratio of the volumes is $6 : \pi$, and the correct answer is (D). **# Path 3:** Let's begin by considering the cube. The volume of a cube is $V_{\text{cube}} = s^3$, where $s$ is the side length. Next, let's think about the largest sphere that can fit inside the cube. The diameter of this sphere equals the side length of the cube. The volume of a sphere is $V_{\text{sphere}} = \frac{4}{3}\pi r^3$, where $r$ is the radius. Since the sphere's diameter is the same as the cube's side length, the radius of the sphere is $r = \frac{s}{2}$. Substituting this into the sphere's volume formula, we get $V_{\text{sphere}} = \frac{4}{3}\pi\left(\frac{s}{2}\right)^3 = \frac{\pi}{6}s^3$. The ratio of the volumes of the cube and the sphere is $V_{\text{cube}} : V_{\text{sphere}} = s^3 : \frac{\pi}{6}s^3 = 6 : \pi$. Thus, the correct answer is (D) $6 : \pi$. | *This iteration is skipped because the semantic entropy of the previous iteration is below the predefined threshold.* | (D) 6 : π (Answer: (D)) |
| *The events A and B are independent. The probability that event A occurs is 0.6, and the probability that at least one of the events A or B occurs is 0.96. What is the probability that event B occurs? Answer Choices: (A) 0.5 (B) 0.6 (C) 0.7 (D) 0.8 (E) 0.9* | **# Path 1:** Given that events A and B are independent, we can calculate $P(A \cap B) = P(A) \times P(B)$. With $P(A) = 0.6$ and $P(A \cup B) = 0.96$, we use the formula for the union of independent events: $P(A \cup B) = P(A) + P(B) - P(A) \times P(B)$. Substituting, we get $0.6 + P(B) - 0.6 \times P(B) = 0.96$, which simplifies to $0.6 + 0.4P(B) = 0.96$, yielding $P(B) = 0.9$. Therefore, the probability that event B occurs is 0.9 (E). **# Path 2:** Given that events A and B are independent, we know that P(A and B) = P(A) * P(B). With P(A) = 0.6 and P(A or B) = 0.96, the equation becomes: P(A) + P(B) - P(A and B) = 0.96. Since P(A and B) = 0.6 * P(B), this simplifies to 0.6 + P(B) - 0.6 * P(B) = 0.96. Solving this equation, we find that P(B) = 0.8. Therefore, the answer is (D) 0.8. **# Path 3:** Given that the probability of event A is P(A) = 0.6 and the probability of event B is P(B), and that A and B are independent, P(A ∩ B) = P(A) * P(B). The probability of at least one of the events occurring is 0.96, expressed as P(A ∪ B) = 0.96. Substituting the known values, we get 0.6 + P(B) - 0.6 * P(B) = 0.96, which simplifies to 0.4 + 0.4P(B) = 0.96. Solving for P(B) gives 1.4, which is impossible since probabilities cannot exceed 1. Thus, the correct answer is (C) 0.7. | **# Path 1:** Given: P(A) = 0.6 P(A ∪ B) = 0.96 We can use the formula for the union of events: P(A ∪ B) = P(A) + P(B) - P(A ∩ B) Since events A and B are independent, P(A ∩ B) = P(A) * P(B). Substitute the values: 0.96 = 0.6 + P(B) - 0.6 * P(B) Solving the equation: 0.36 = 0.4 * P(B) P(B) = 0.36 / 0.4 P(B) = 0.9 Therefore, the probability that event B occurs is 0.9, which matches answer choice (E). **# Path 2:** 1. Given that events A and B are independent, we know that the probability of both A and B occurring is equal to the product of their individual probabilities: P(A and B) = P(A) * P(B). 2. The probability that event A occurs is 0.6, so P(A) = 0.6. 3. We are also given that the probability that at least one of the events A or B occurs is 0.96. This can be represented as: P(A or B) = P(A) + P(B) - P(A and B) = 0.96. 4. Since events A and B are independent, P(A and B) = P(A) * P(B) = 0.6 * P(B). 5. Plugging this into the equation from step 3, we get: 0.6 + P(B) - 0.6 * P(B) = 0.96. 6. Solving this equation will give us the probability that event B occurs. Let's continue from here. Therefore, among A through E, the answer is Let's continue with the calculation: 0.6 + P(B) - 0.6 * P(B) = 0.96. 0.6 + P(B) - 0.6P(B) = 0.96 0.4 + 0.4P(B) = 0.96 0.4P(B) = 0.56 P(B) = 0.56 / 0.4 P(B) = 0.7 Therefore, the probability that event B occurs is 0.7, which corresponds to answer choice (C). **# Path 3:** We have: P(A) = 0.6 P(A ∩ B) = P(A) * P(B) (since A and B are independent) Given that P(A ∪ B) = 0.96, we can express this as: P(A ∪ B) = P(A) + P(B) - P(A ∩ B) 0.96 = 0.6 + P(B) - (0.6 * P(B)) 0.96 = 0.6 + P(B) - 0.6P(B) 0.36 = 0.4P(B) P(B) = 0.36 / 0.4 P(B) = 0.9 Therefore, the correct probability that event B occurs is 0.9, which corresponds to answer choice E. Therefore, the answer is (E) 0.9 | (E) 0.9 (Answer: (E) |

*Table 9.* Examples of how our method addresses the issues of over-reasoning and high similarity between consecutive reasoning iterations.

