# OpenReview forum: "Rethinking Chain-of-Thought from the Perspective of Self-Training"
_ICML.cc/2025/Conference — ICML 2025 poster_

### Official Review · Reviewer_Pmva · 2025-03-06

**Overall Recommendation:** 4

**Summary:**

This paper investigates CoT approach and proposes a novel CoT framework inspired by its similarity to self-training to enhance reasoning performance. The framework consists of two core modules: a task-specific prompt module that optimizes the initial reasoning process and an adaptive reasoning iteration module that dynamically adjusts the reasoning procedure to address issues in existing CoT methods, such as over-reasoning and high similarity across consecutive reasoning iterations. Experimental results demonstrate that the proposed approach achieves significant improvements in both performance and computational efficiency.

**Claims And Evidence:**

The key claims of this paper include: (1) CoT reasoning and self-training share the same core objective; (2) reasoning performance can be improved through task-specific prompting and adaptive reasoning iteration; (3) the proposed approach outperforms traditional CoT methods in computational efficiency. Experimental results support these claims, demonstrating improvements in both performance and computational efficiency.

**Essential References Not Discussed:**

None

**Experimental Designs Or Analyses:**

The paper validates the effectiveness of the proposed method through extensive experiments, focusing primarily on the improvements in reasoning performance and computational efficiency.

**Methods And Evaluation Criteria:**

The proposed approach effectively integrates task-specific prompt optimization and adaptive reasoning iteration, enhancing reasoning capabilities while reducing computational overhead. The evaluation criteria employed in this paper align with those of traditional CoT methods, wherein key segments of the generated output are extracted and compared against the ground truth.

**Other Comments Or Suggestions:**

1)Is it better to have a larger maximum number of iterations in the proposed ARI module?
2)This paper should include an intuitive analysis of the prompts selected by the TSP, such as whether they provide stronger guidance or how they structurally differ from manually crafted prompts.

**Other Strengths And Weaknesses:**

Strengths：
1) This paper is grounded in solid theoretical foundations, providing an in-depth theoretical analysis of entropy variation. Additionally, the experimental results are significant, demonstrating that the proposed framework outperforms baseline methods in both performance and computational efficiency, particularly excelling in arithmetic reasoning tasks.

2) In Section 2, this paper provides an in-depth theoretical discussion on the uncertainty measure (information entropy) in self-training, analyzing the mechanism by which the iterative generation of pseudo-labels in semi-supervised self-training leads to progressively improved prediction accuracy. Subsequently, the authors apply this "entropy reduction" perspective to CoT reasoning, and in the following experiments, they conduct a quantitative comparison using semantic entropy, supported by both theoretical analysis and extensive experimental validation.

3) The Task-Specific Prompt module proposed by the authors automatically searches for or constructs prompt statements from task samples, rather than using the generic "Let's think step by step." In scenarios that require more precise guidance, this approach may yield better results. The authors' experiments also confirm that targeted prompts can significantly improve the results of the first-round reasoning.

4) The Adaptive Reasoning Iteration module proposed by the authors incorporates a semantic entropy-based exit mechanism, preventing further reasoning when the model is already "highly confident," thus avoiding the introduction of additional errors. Additionally, to reduce excessive repetition of reasoning across different iterations and control reasoning similarity, the module introduces new prompts. This encourages LLMs to explore more diverse reasoning paths.

5) The overall logic is clear, and the writing is smooth.

Weaknesses:
1) There is a lack of discussion on negative results. While the paper primarily demonstrates the advantages of the method, it does not provide an explanation for the model's poor performance on certain datasets, such as why the improvement in the "commonsense" task is limited.

2) The main text lacks a Related Work section, and the Task-Specific Prompt part of Framework Figure 4 does not clearly describe the specific details, making it difficult for readers to understand how to obtain the candidate prompts and select the optimal one.

**Questions For Authors:**

Does the number of self-consistency samples have any particular impact on the proposed method?

**Relation To Broader Scientific Literature:**

This study builds upon CoT reasoning and self-training methods, proposing a new framework that combines the advantages of both. Compared to existing CoT reasoning methods, the proposed approach introduces task-specific prompt optimization and adaptive reasoning iteration, thereby reducing issues of over-reasoning and high similarity.

**Theoretical Claims:**

The theoretical contribution of the paper primarily focuses on the analysis of entropy variation in self-training. The claims are intuitively reasonable, and the proofs provided in the Appendix should be correct.

---

> ### Author Rebuttal · Authors · 2025-03-30
>
> We sincerely thank the reviewer for the valuable feedback and encouraging comments. We are motivated by the suggestions and have addressed each concern in detail as follows:
>
> > **Q1.** There is a lack of discussion on negative results. While the paper primarily demonstrates the advantages of the method, it does not provide an explanation for the model's poor performance on certain datasets, such as why the improvement in the "commonsense" task is limited.
>
>  **A1.** We acknowledge that the performance gain is relatively limited on commonsense tasks. This is mainly because such tasks rely more on pretrained world knowledge and co-occurrence patterns, rather than explicit logical steps. In contrast, arithmetic tasks better align with our iterative reasoning framework. For open-ended tasks, additional reasoning may introduce noise rather than benefit. We will explore better adaptations to such tasks in future work.
>
> > **Q2.** The main text lacks a Related Work section, and the Task-Specific Prompt part of Framework Figure 4 does not clearly describe the specific details, making it difficult for readers to understand how to obtain the candidate prompts and select the optimal one.
>
>  **A2.** In the final version, we will add a “Related Work” section to cover relevant studies on Self-Training, Chain-of-Thought reasoning, and semantic entropy. We will also refine Figure 4 and clarify that the task-specific prompt module generates candidate prompts based on several example questions, evaluates them using average semantic entropy on a validation set, and selects the one with the lowest entropy as the final prompt.
>
> > **Q3.** Is it better to have a larger maximum number of iterations in the proposed ARI module?
>
>  **A3.** We have added a sensitivity analysis of the maximum number of iterations $T$, with the experimental results shown in Table. Overall, increasing the maximum number of iterations does not always lead to better performance. Taking AQuA and AddSub as examples, the model already achieves a relatively high accuracy at $T=3$, while further increasing $T$ to 4 or 5 leads to saturated or slightly fluctuating performance, along with a significant increase in computational cost. At the same time, for more challenging tasks, a larger maximum iteration count may offer a broader reasoning space and thus yield further performance improvements. Therefore, the optimal value of $T$ should be determined based on task difficulty: for simpler tasks, a smaller $T$ is more efficient; whereas for more complex tasks, allowing a larger $T$ may better support sufficient reasoning.
>
> | Dataset| $T=1$| $T=2$| $T=3$| $T=4$| $T=5$|
> |-|-|-|-|-|-|
> | AQuA     | 59.5%| 70.8%| 70.1%| 71.7%| 71.3%|
> | AddSub   | 80.5%| 86.1%| 88.4%| 88.8%| 88.6%|
>
> > **Q4.** This paper should include an intuitive analysis of the prompts selected by the TSP, such as whether they provide stronger guidance or how they structurally differ from manually crafted prompts.
>
>  **A4.** Our proposed TSP module is designed to automatically generate initial reasoning prompts that better align with the task’s semantic distribution, serving as a replacement for generic prompts. While generic prompts possess a certain degree of generality, their expressions tend to be overly abstract and fail to effectively guide the model to focus on the key reasoning elements within a task. In contrast, TSP leverages actual task samples to generate multiple candidate prompts and selects the most effective one based on average semantic entropy, thereby improving the quality of initial reasoning and reducing the number of iterations. Structurally, prompts generated by TSP typically extend the generic prompt template with additional task-specific semantic cues. For example, for the AQuA dataset, TSP produces a prompt: "Let's think step by step, how to break down the mathematical operations involved in each problem and identify the key concepts to solve them accurately".
>
> > **Q5.** Does the number of self-consistency samples have any particular impact on the proposed method?
>
>  **A5.** We conducted additional experiments on four datasets to evaluate the impact of the sampling number $N$ on model performance and robustness, as shown in Table below. The results show that as $N$ increases, the overall model accuracy improves, with more pronounced gains on challenging tasks such as AMC2023 and Gaokao. Meanwhile, performance tends to saturate after $N=3$, indicating that even a relatively small $N$ can yield stable semantic entropy estimates. This balances performance with computational efficiency and demonstrates the method’s strong robustness with respect to the parameter $N$.
>
> | Dataset| $N=1$| $N=2$| $N=3$| $N=4$| $N=5$|
> |-|-|-|---------|---------|---------|
> | AQuA      | 53.9%  | 70.9%  | 70.1%  | 71.3%  | 72.4%|
> | AddSub    | 73.9%  | 85.1% | 88.4%  | 88.9%  | 90.4%|
> | AMC2023   | 12.5%  | 20.0%  | 25.0%  | 27.5%  | 30.0%|
> | Gaokao    | 32.8%  | 41.3%  | 44.2%  | 44.7%  | 45.3%|

---

### Official Review · Reviewer_XjxR · 2025-03-12

**Overall Recommendation:** 3

**Summary:**

Inspired by self-training, this paper designs the CoT framework to improve reasoning performance. It contains two core elements: a task-specific prompt and an adaptive reasoning iteration. This paper finally conducted experiments on 10 reasoning datasets and achieved improved results.

**Claims And Evidence:**

I think the discussion on Semantic Entropy for LLM reasoning based on self-training is interesting, but it lacks theoretical proof and experimental verification in more scenarios.

**Essential References Not Discussed:**

Overall complete.

**Experimental Designs Or Analyses:**

See details in Methods And Evaluation Criteria.

**Methods And Evaluation Criteria:**

I think the main experiment needs to not only compare the effects, but also the token costs brought by the comparison methods.

**Other Comments Or Suggestions:**

There are some minor typos in the writing of the paper, for example, the first letter of "we" in line 105 should be capitalized, etc.

**Other Strengths And Weaknesses:**

I think the design of the idea from self-training to semantic entropy is interesting. However, I doubt whether this method will bring additional token overhead, and the effect in the experiment does not seem to be very significant. In addition, the effectiveness of semantic entropy is mainly verified from experimental phenomena, and I have concerns about its adaptability and generalization ability in more scenarios. Furthermore, the assumption of a mixture of Gaussians in the proof seems overly ideal.

**Questions For Authors:**

1. How to determine the optimal predefined threshold  $\delta$? 2. Does this pipeline have a certain degree of generalization ability? For example, is the optimized COT adaptable to new tasks? Or does it mean that each task needs to be optimized in a specific way?

**Relation To Broader Scientific Literature:**

The optimization and design of chain of thought may be helpful for the automated design of agents and the generalization research of ML.

**Theoretical Claims:**

Lack of sufficient theoretical claims on LLM Semantic Entropy.

---

> ### Author Rebuttal · Authors · 2025-03-30
>
> We sincerely thank the reviewer for the valuable feedback and encouraging comments. We are motivated by the suggestions and have addressed each concern in detail as follows:
>
> > **Q1.** Main experiments should compare both effectiveness and token overhead of each method.
>
> **A1.** We have added token consumption to the main experiments to assess performance-cost trade-offs. Our method outperforms others while using fewer tokens than the SOTA method (e.g., Nash), especially on challenging tasks like AMC2023 and Gaokao, showing better efficiency in complex reasoning.
>
> | Method|AQuA|Token|AddSub|Token|AMC2023|Token|Gaokao|Token|
> |-|-|-|-|-|-|-|-|-|
> |Zero|39.8%|614|57.5%|117|2.5%|841|31.3%|508|
> |SC|63.0%|1,875|81.5%|743|15.0%|2,950|35.9%|3,357|
> |Active|66.1%|1,649|86.3%|542|12.5%|2,421|40.5%|3,409|
> |Nash|68.9%|6,598|88.9%|4,431|25.0%|12,940|43.0%|8,705|
> |Our|70.1%|2,884| 89.1%|1,028|27.5%|7,604|44.2%|4,069|
>
> > **Q2.** Lack of sufficient theoretical claims on LLM Semantic Entropy.
>
> **A2.** The highly nonlinear nature of LLMs makes it difficult to model their output distribution using traditional probabilistic methods (e.g., GMM), posing challenges for establishing a rigorous theoretical framework for semantic entropy. This paper focuses on proposing a practical, scenario-driven CoT optimization framework, using semantic entropy as a heuristic metric to capture answer distribution dynamics during reasoning. While its theoretical foundation is not fully established, experiments demonstrate its effectiveness, and we will further investigate its theoretical basis in future work.
>
> > **Q3.** I have concerns about its adaptability and generalization ability in more scenarios.
>
> **A3.** To evaluate the generalization of the semantic entropy mechanism across tasks, we added experiments on two distinct datasets: Race (English reading comprehension task) and Anli (adversarial natural language inference task), each with 300 test samples. As shown in Table, our method achieved clear performance gains on both, demonstrating strong robustness across language understanding tasks. We plan to explore its application to Vision and Cross-Modal tasks in future work.
>
> | Method|Race|Token|Anli|Token|
> |-|-|-|-|-|
> |Zero|80.0%|508|45.0%|268|
> |SC|84.5%|2,530|56.0%|1,456|
> |Our|88.5%|3,467|64.5%|2,245|
>
> > **Q4.** The assumption of a mixture of Gaussians in the proof seems overly ideal.
>
> **A4.** We adopt GMM as the modeling assumption based on two reasons: (1) GMM is a widely used and interpretable model that helps illustrate how pseudo-labels reduce entropy; (2) our results are a direct corollary of [1], which provides a general derivation under the sub-exponential family, of which GMM is a special case. To balance rigor and clarity, we use GMM as a heuristic example, while the conclusions also hold under broader distributions given certain conditions.
>
> [1] Frei S,et al. self-training converts weak learners to strong learners in mixture models.
>
> > **Q5.** How to determine the optimal predefined threshold $\delta$?
>
> **A5.** The predefined threshold $\delta$ is empirically determined based on the task type and the number of samples $N$ in self-consistency (SC). Specifically, we allow up to $k$ predictions to deviate from the majority class and compute the corresponding entropy threshold using the ratio $(N - k)/N$. For example, in the more challenging reasoning task AQuA, when $N = 4$ and $k = 1$ (i.e., allowing at most 1 out of 4 predictions to differ from the majority), the corresponding threshold is calculated as:
> $\delta = -\left( \frac{N-k}{N} \log \frac{N-k}{N} + \frac{k}{N} \log \frac{k}{N} \right) \approx 0.811$.
>
> As shown in Table, setting the threshold too high may cause LLM to terminate early before identifying the correct answer, thus harming accuracy. Conversely, setting it too low may lead to excessive reasoning and the introduction of noise, which can also degrade performance. Therefore, we typically set $k$ between 0 and 2, and use it to compute semantic entropy as the decision threshold $\delta$. As $N$ increases, we appropriately raise $k$ to enhance the robustness of the decision-making process.
>
> |Allowed Inconsis / SC|$N=2$|$N=3$|$N=4$|$N=5$|
> |-|-|-|-|-|
> |$k=0$|70.9%|70.1%|71.3%|72.4%|
> |$k=1$|--|68.2%|72.8%|73.6%|
> |$k=2$|--|--|68.7%|71.7%|
> |$k=3$|--|--|--|69.3%|
>
> > **Q6.** Does the pipeline generalize across tasks, or must the optimized CoT be re-tuned for each one?
>
> **A6.** Our framework includes a reasoning pipeline composed of the TSP and ARI modules. While ARI is task-agnostic and requires no adaptation, TSP involves optimization on a single source task. Once optimized, the entire pipeline can be directly applied to other tasks of a similar nature without re-tuning. This demonstrates the generalizability of our pipeline across tasks.
>
> |Source / Target|AQuA|AddSub|AMC2023|Gaokao|
> |-|-|-|-|-|
> |AQuA|70.1%|85.6%|20.0%|40.2%|
> |AddSub|67.7%|88.4%|20.0%|41.3%|
> |AMC2023|68.1%|86.3%|25.0%|41.9%|
> |Gaokao|66.9%| 84.3%|17.5%|44.2%|

---

### Official Review · Reviewer_wWvP · 2025-03-12

**Overall Recommendation:** 4

**Summary:**

This paper explores the conceptual similarity between CoT reasoning and self-training, highlighting their shared goal of minimizing predictive uncertainty by iteratively leveraging model-generated information. Based on this insight, the authors propose a novel CoT framework integrating a Task-Specific Prompt module and an Adaptive Reasoning Iteration module. Experiments on 10 reasoning datasets demonstrate that the proposed approach significantly outperforms baseline CoT methods in both reasoning performance and computational efficiency, with particularly strong results on arithmetic datasets. The main contributions include establishing a connection between CoT reasoning and self-training through entropy minimization, introducing TSP and ARI modules, and achieving notable improvements in complex reasoning tasks with strong generalization and efficiency.

**Claims And Evidence:**

The proposed method is generally clear and compelling. The authors support their main contributions with experimental results on three types of reasoning tasks and corresponding theoretical analysis.

**Essential References Not Discussed:**

None

**Experimental Designs Or Analyses:**

The experimental design effectively evaluates the proposed method. The results show that the method achieves strong performance on multiple benchmark datasets, and both experiments with fixed iteration counts and comparison experiments support the authors' main conclusions.

**Methods And Evaluation Criteria:**

The proposed method is well-founded and suitable for the research problem. The evaluation criteria align with existing benchmarks, and the experimental setup is reasonable.

**Other Comments Or Suggestions:**

The symbol definitions in the formulas are not clear.

**Other Strengths And Weaknesses:**

Strengths:

1.	The paper presents an interesting and insightful perspective by drawing an analogy between CoT reasoning and self-training, especially the pseudo-labeling strategy, highlighting their common goal of "iteratively reducing predictive uncertainty."

2.	The two modules proposed in the paper (TSP and ARI) can serve as plug-and-play enhancement components.

3.	The writing is clear. The authors provide theoretical proofs, algorithm pseudocode, and examples of reasoning processes in the appendix.

4.	The paper selects datasets that cover various types of reasoning, demonstrating the method's versatility.


Weaknesses:

1. Although the authors propose a method for automatically searching for the "optimal prompt," they do not provide specific examples in the experiments, nor do they analyze why it outperforms general prompts (e.g., "let's think step by step").

2. Some technical details lack contextual background. For instance, the "Jaccard Index" is mentioned directly in the method section, but its role or the rationale for choosing this metric is not explained.

3. There is no sensitivity analysis of the maximum number of iterations.

**Questions For Authors:**

Why does the proposed method perform better on arithmetic datasets than on commonsense datasets?

**Relation To Broader Scientific Literature:**

The paper is consistent with research in the related field and provides new insights into the essence of chain-of-thought.

**Theoretical Claims:**

The theoretical arguments presented in the paper have been carefully examined and are generally correct.

---

> ### Author Rebuttal · Authors · 2025-03-30
>
> We sincerely thank the reviewer for the valuable feedback and encouraging comments. We are motivated by the suggestions and have addressed each concern in detail as follows:
>
> > **Q1.** Although the authors propose a method for automatically searching for the "optimal prompt," they do not provide specific examples in the experiments, nor do they analyze why it outperforms general prompts (e.g., "let's think step by step").
>
> **A1.** Our proposed TSP aims to automatically generate initial reasoning prompts that better align with the specific task’s semantic distribution, serving as a replacement for generic prompts. Although generic prompts exhibit a certain degree of adaptability across most tasks, their expressions are relatively generalized and often fail to sufficiently guide LLMs to focus on the critical reasoning paths required by the task. The TSP module leverages real task samples to guide the LLM in generating multiple candidate prompts, then selects the optimal one based on the evaluation metric of average semantic entropy. This enhances the quality of initial reasoning and reduces the number of iterations. Below are examples of optimal prompts automatically searched on different datasets, showing how they further integrate task-specific features on top of the generic template:
>
> - AQuA: "Let's think step by step, how to break down the mathematical operations involved in each problem and identify the key concepts to solve them accurately."
> - AddSub: "Let's think step by step, how to efficiently solve these word problems involving basic arithmetic operations and simple logic."
>
> > **Q2.** Some technical details lack contextual background. For instance, the "Jaccard Index" is mentioned directly in the method section, but its role or the rationale for choosing this metric is not explained.
>
> **A2.** We introduce the Jaccard Index in the method section to measure the lexical-level similarity between reasoning results of adjacent iterations, aiming to determine whether the current iteration has generated sufficiently diverse reasoning paths, thereby avoiding repetitive or redundant reasoning by the model. The Jaccard Index is chosen for its simplicity and efficiency in evaluating the overlap between two sets, making it particularly suitable for quickly assessing word-level differences between texts and effectively capturing trends in reasoning diversity. We will supplement the relevant background explanation in the paper to enhance readability and completeness.
>
> > **Q3.** There is no sensitivity analysis of the maximum number of iterations.
>
> **A3.** We have added a sensitivity analysis of the maximum number of iterations $T$, with the experimental results shown in Table. It can be observed that the model achieves a significant accuracy improvement at $T=3$, and the performance tends to saturate or even fluctuate slightly after $T=4$, indicating that a relatively small number of iterations is sufficient to achieve near-optimal reasoning performance. Taking the AQuA and AddSub datasets as examples, the accuracy improvement slows down notably after $T=3$, suggesting limited gains from further iterations. Combined with our proposed semantic entropy-based early stopping mechanism, the system can dynamically decide whether to continue reasoning for each sample, effectively reducing unnecessary computational overhead while maintaining performance.
>
> | Dataset   | $T=1$      | $T=2$      | $T=3$      | $T=4$      | $T=5$      |
> |----------|--------|--------|--------|--------|--------|
> | AQuA     | 59.5% | 70.8% | 70.1% | 71.7% | 71.3% |
> | AddSub   | 80.5% | 86.1% | 88.4% | 88.8% | 88.6% |
>
>
>
> > **Q4.** The symbol definitions in the formulas are not clear.
>
> **A4.** Thank you for the comment. We have carefully double-checked the paper and revised the notations. In the final version, we will ensure that all symbols are clearly defined at first use.
>
> > **Q5.** Why does the proposed method perform better on arithmetic datasets than on commonsense datasets?
>
> **A5.** We believe this difference primarily stems from the varying degrees to which different task types rely on the capabilities of language models. Arithmetic datasets emphasize strict logical steps and step-by-step calculation processes, which closely align with our method’s design that involves multi-turn reasoning and explicit intermediate steps. The model can iteratively approach the correct answer, thus benefiting significantly. In contrast, commonsense tasks rely more on the world knowledge and language co-occurrence patterns acquired during pretraining. These tasks have a more open reasoning space, more diverse answer forms, and often lack a clear logical chain. As a result, multi-turn reasoning offers limited gains for such tasks and may even introduce unnecessary redundant information that affects the final judgment.

---

### Official Review · Reviewer_Yf9H · 2025-03-18

**Overall Recommendation:** 3

**Summary:**

This paper discusses the similarities between Chain-of-Thought (CoT) reasoning and self-training, and proposes how to reduce prediction uncertainty for both iteratively leveraging on model-generated information. In particular, this paper introduces a novel CoT framework with two main ingredients: (i) a task-specific prompt module designed to optimize the initial reasoning process by searching for prompts that minimize uncertainty, and (ii) an adaptive reasoning iteration module that dynamically refines the reasoning process to address over-reasoning and similarity between consecutive iterations. Based on theoretical results from self-training, the authors link entropy variations and semantic entropy to CoT. Experimental results to demonstrate the effectiveness of their proposed framework in improving reasoning performance and computational efficiency across several datasets.

**Claims And Evidence:**

The paper claims that CoT reasoning shares the core objective of uncertainty reduction with self-training and that their proposed framework, inspired by this analogy, improves CoT performance. While the empirical evidence presented in the experiments generally supports the performance improvements of the proposed framework, the theoretical link between self-training's entropy variation and CoT reasoning is more conceptual and analogical than rigorously derived. The claim that semantic entropy is an effective metric for guiding CoT iterations and prompt selection is supported by experimental results, but lacks direct comparative evidence against more conventional prompt selection methods. The justification for the specific design choices, such as the exploratory prompt and the adaptive iteration strategy, relies more on intuitive reasoning and analogy to self-training principles than on strong theoretical backing specific to CoT.

**Essential References Not Discussed:**

To my best knowledge, this paper includes most of the essential references in the Introduction section.

However, in order to improve the readability, the paper should improve the discussion of essential references. The lack of "Related Work" section leads to insufficient discussions of the proposed work in the context of existing related ones.

For example, the paper should discuss and ideally compare against more conventional prompt selection methods used in the CoT literature, such as those relying on simpler heuristics or ensemble-based prompt selection. Additionally, while the paper cites some popular CoT methods as baselines, including more recent and stronger CoT baselines in the experimental comparison would provide a more comprehensive evaluation of the proposed framework's advancements.

**Experimental Designs Or Analyses:**

The experimental designs and analyses are generally sound and follow standard practices for evaluating CoT methods. The ablation studies effectively demonstrate the contribution of the task-specific prompt and adaptive reasoning iteration modules.

However, a notable weakness is the lack of ablation studies on the number of self-consistency samples (N), which is a crucial parameter influencing semantic entropy calculation and the overall framework's effectiveness.

Furthermore, the experimental section would be strengthened by including a direct comparison of the proposed semantic entropy-based prompt selection method with more conventional prompt selection techniques to justify its added complexity and computational cost.

**Methods And Evaluation Criteria:**

The proposed methods, including the task-specific prompt module using semantic entropy for prompt selection and the adaptive reasoning iteration module with entropy-based stopping and exploratory prompts, are novel and relevant to addressing limitations in traditional CoT approaches. The use of semantic entropy as a metric for uncertainty in CoT reasoning is an interesting approach. The evaluation criteria, using standard benchmark datasets across arithmetic, commonsense, and symbolic reasoning, are generally appropriate for evaluating CoT methods. However, the benchmarks used are not the most challenging and cutting-edge datasets in these domains.

**Other Comments Or Suggestions:**

* Consider strengthening the theoretical justification for applying self-training-inspired entropy principles to CoT reasoning.
* Include experiments comparing the semantic entropy-based prompt selection with simpler prompt selection methods.
* Perform ablation studies on the number of self-consistency samples (N) to assess its impact.
* Evaluate the framework on more challenging and cutting-edge benchmarks, especially in mathematical reasoning.
* Expand the baseline comparison to include more recent and stronger CoT methods.
* Clarify the time cost analysis and the specific mechanisms through which adaptive iteration achieves time efficiency gains.

**Other Strengths And Weaknesses:**

Strengths:
* Originality: The paper presents a novel perspective on CoT reasoning by drawing inspiration from self-training and introducing semantic entropy as a guiding metric.
* Significance: The proposed framework offers a potentially effective approach to improve CoT performance and address limitations like over-reasoning and lack of diversity in reasoning paths.
* Clarity: The paper is generally well-written and clearly explains the proposed framework and experimental setup.
* Empirical Validation: The experimental results demonstrate promising performance improvements across various datasets.

Weaknesses:
* Conceptual Link Strength: The analogy between self-training and CoT, while interesting, could be theoretically strengthened. The justification for transferring entropy-based principles from self-training to CoT reasoning is not fully rigorous.
* Computational Cost: The computational cost of task-specific prompt selection using semantic entropy and the per-iteration cost of adaptive reasoning could be a concern, and needs more thorough justification and comparison to simpler alternatives.
* Limited Baselines/Benchmarks: The choice of baselines and benchmarks could be more comprehensive and challenging to fully showcase the advantages of the proposed framework.
* Theoretical Justification for Exploratory Prompt: The theoretical grounding for the specific exploratory prompt design in adaptive iteration could be more explicitly linked to the identified limitations of CoT.
* Unclear Time Efficiency Benefit: The claimed time efficiency gains of adaptive iteration compared to fixed iteration are not entirely clear from the provided analysis.

**Questions For Authors:**

1. Theoretical Justification of Self-Training Analogy: While the analogy to self-training is interesting, could the authors elaborate on the theoretical justification for directly applying entropy-based principles from self-training (which involves parameter updates) to CoT reasoning (which manipulates inputs without changing model weights)? How can Theorem 2.2, derived for self-training, be rigorously argued to motivate the design choices in the proposed CoT framework? A more detailed explanation of this theoretical bridge would strengthen the paper.
2. Comparison to Conventional Prompt Selection: The task-specific prompt module relies on computationally expensive semantic entropy calculations. Could the authors include experiments comparing this approach to more conventional and computationally cheaper prompt selection methods (e.g., simpler heuristics, frequency-based methods) in terms of both performance and computational efficiency? Demonstrating a clear advantage over simpler methods would better justify the added complexity.
3. Theoretical Basis for Exploratory Prompt: The exploratory prompt "reevaluate from alternative perspectives" is designed to encourage diversity. Could the authors provide a more explicit theoretical justification for why this specific prompt design is expected to effectively promote exploration and reduce semantic entropy in subsequent CoT iterations, especially in relation to the issue of high similarity between consecutive iterations identified in Section 2.2?
4. Ablation Study on Self-Consistency Sampling (N): The number of self-consistency samples (N) is a crucial parameter for semantic entropy estimation. Could the authors include ablation studies varying N to analyze its impact on the framework's performance and robustness? Understanding the sensitivity to N is important for practical application and evaluating the reliability of the semantic entropy metric.
5. Rationale for Baseline Choices and Benchmark Complexity: What was the rationale behind choosing Contrastive-CoT and RE2 as the primary baselines? Were more recent and potentially stronger CoT methods considered? Furthermore, while the benchmarks are standard, could the authors discuss why these benchmarks were chosen and whether evaluating on more challenging, cutting-edge datasets, particularly in mathematical reasoning, could further validate the framework's capabilities and significance?
6. Clarification of Time Efficiency in Adaptive Iteration: Figure 5b suggests time efficiency gains with adaptive iteration. Could the authors clarify how the adaptive iteration method achieves this efficiency gain, given that each iteration involves two LLM calls? Is the time reduction primarily due to the early stopping mechanism when semantic entropy is low, or are there other factors contributing to the improved time efficiency compared to fixed iteration? A clearer explanation of the trade-offs between per-iteration cost and total iterations in adaptive reasoning would be valuable.

**Relation To Broader Scientific Literature:**

The paper builds upon the established literature of chain-of-thought reasoning and connects it to the well-studied field of self-training. The key contribution lies in applying the concept of uncertainty reduction, inspired by self-training, to enhance CoT reasoning through semantic entropy-guided prompt selection and adaptive iteration. The use of semantic entropy as a metric for uncertainty in CoT and the proposed adaptive iteration strategy are novel contributions to the CoT literature.

**Theoretical Claims:**

The paper includes theoretical claims regarding entropy variation in self-training (Lemma 2.1 and Theorem 2.2). While I did not thoroughly check the correctness of the proofs, the paper primarily relies on referencing existing theoretical results in self-training. It's important to note that these theoretical claims directly apply to self-training and are used to motivate and inspire the design of the CoT framework, rather than providing a rigorous theoretical foundation for the proposed CoT method itself. The link between self-training's entropy theorems and the proposed CoT framework is thus analogical and motivational, not deductively derived.

---

> ### Author Rebuttal · Authors · 2025-03-30
>
> We sincerely thank the reviewer for the valuable feedback and encouraging comments. We are motivated by the suggestions and have addressed each concern in detail as follows:
>
> > **Q1.** The paper lacks sensitivity analysis for the sampling number $N$.
>
> **A1.** We evaluated the impact of $N$ on four datasets. Accuracy improves with larger $N$, especially on harder tasks, but saturates after $N=3$, suggesting small $N$ is sufficient for stable semantic entropy estimation.
>
> |Dataset|$N=1$|$N=2$|$N=3$|$N=4$|$N=5$|
> |-|-|-|-|-|-|
> |AQuA|53.9%|70.9%|70.1%|71.3%|72.4%|
> |AddSub|73.9%|85.1%|88.4%|88.9%|90.4%|
> |AMC2023|12.5%|20.0%|25.0%|27.5%|30.0%|
> |Gaokao|32.8%|41.3%|44.2%|44.7%|45.3%|
>
> > **Q2.** TSP incurs extra cost. Can the authors empirically show its advantage in both performance and efficiency over traditional prompt selection?
>
> **A2.** We compared prompt selection methods, including traditional ensemble prompting (generates multiple prompts and aggregates outputs) and iterative search (automatically generates and filters prompts using Monte Carlo search). These were compared with the TSP module based on semantic entropy. TSP achieves the highest accuracy across all datasets while significantly reducing total token consumption (prompt search $S_{\text{Token}}$ + inference $T_{\text{Token}}$). Compared to the search method, TSP balances performance and efficiency better.
>
> |Method|AQuA|$S_{\text{Token}}$+$T_{\text{Token}}$|AddSub|$S_{\text{Token}}$+$T_{\text{Token}}$|AMC2023|$S_{\text{Token}}$+$T_{\text{Token}}$|Gaokao|$S_{\text{Token}}$+$T_{\text{Token}}$|
> |-|-|-|-|-|-|-|-|-|
> |Ensemble|59.7%|1,703+5,957|80.4%|790+2,196|12.5%|2,089+7,749|33.6%|2,234+7,310|
> |Search|64.2%|1,932+53,581|84.4%|918+30,732|12.5%|2,672+67,642|37.6%|2,906+81,109|
> |TSP|66.1%|2,050+21,570|85.8%|1,005+10,281|17.5%|2,261+24,506|41.9%|2,554+23,716|
>
> > **Q3.** How can the entropy principle from self-training, which involves parameter updates, be applied to CoT that only modifies inputs?
>
> **A3.** We attempt to draw an analogy between the principle of self-training and CoT. Although they differ in operational mechanisms, with self-training relying on parameter updates and CoT guiding reasoning through input manipulation, we argue that both exhibit structural consistency in terms of information compression and entropy convergence. In CoT, while model parameters remain unchanged, the iterative optimization of the reasoning path set $R_t$ gradually leads to semantic-level distributional convergence over the reasoning space $\mathcal{R}$. As iterations progress, the proportion of reasoning paths containing correct subpaths $R'$ increases, and these paths tend to converge toward the correct answer cluster. This process results in a gradual increase in the posterior probability of the correct answer, reflecting a reduction in semantic entropy that parallels the entropy minimization objective in self-training.
>
> > **Q4.** How does "re-evaluate from another perspective" reduce reasoning similarity and entropy?
>
> **A4.** The exploratory prompt encourages LLM to break away from the current local reasoning trajectory and actively explore subspaces within the generation space $\mathcal{R} \times \mathcal{A}$ that are semantically orthogonal to the original path. Such orthogonality is reinforced through key semantic constraints embedded within the prompt ("alternative perspectives"), prompting substantial strategic changes in the reasoning process. This process can further converge the answer space, thereby compressing the entropy of the overall candidate answer set.
>
> > **Q5.** Include stronger CoT baselines, harder datasets, and overhead comparison.
>
> **A5.** We have newly incorporated two recent and stronger CoT methods, Active[1] and Nash[2], along with more challenging datasets, AMC2023 and Gaokao. Our method achieves higher accuracy while maintaining reasonable token consumption.
>
> |Method|AQuA|Token|AddSub|Token|AMC2023|Token|Gaokao|Token|
> |-|-|-|-|-|-|-|-|-|
> |Zero|39.8%|614|57.5%|117|2.5%|841|31.3%|508|
> |SC|63.0%|1,875|81.5%|743|15.0%|2,950|35.9%|3,357|
> |Active|66.1%|1,649|86.3%|542|12.5%|2,421|40.5%|3,409|
> |Nash|68.9%|6,598|88.9%|4,431|25.0%|12,940|43.0%|8,705|
> |Our|70.1%|2,884|89.1%|1,028|27.5%|7,604|44.2%|4,069|
>
> [1] Diao S,et al. Active Prompting with Chain-of-Thought for LLMs. \
> [2] Zhang Z,et al. Multi-Path Inference with Preference Equilibrium.
>
> > **Q6.** Adaptive iteration improves efficiency, but the underlying mechanism remains unclear.
>
> **A6.** Adaptive iteration enhances efficiency via early stopping based on semantic entropy. When LLM shows high confidence, it dynamically halts to save computation. As shown in Table, our method averages 2.1 iterations (vs. fixed 3), with 55.9% of samples stopped early, reducing token while maintaining performance.
>
> |Maximum Iterations|1-1(0.0%)|2-1.6(40.2%)|3-2.1(55.9%)|4-2.5(60.1%)|5-3.1(66.5%)|
> |-|-|-|-|-|-|
> |Fixed Iteration|2,154|4,629|6,509|8,717|11,232|
> |Adaptive Iteration|2,154|3,779|4,591|5,458|6,609|

---

> > ### Comment · Reviewer_Yf9H · 2025-04-09
> >
> > Thanks for the authors' responses. This clarifies most of my concerns and very helpful. I'll maintain my scores, learning towards accept.

---

### Decision · Program_Chairs · 2025-05-01

**Decision:**

Accept (poster)

**Comment:**

This paper connects CoT reasoning with self-training, showing both aim to reduce predictive uncertainty using model-generated outputs. Based on this, the authors propose a new CoT framework with a task-specific prompt and adaptive reasoning iteration module. Experiments on several reasoning benchmarks show notable gains in accuracy and efficiency, especially on arithmetic tasks. All reviewers unanimously recommended for acceptance.